# Unveiling the Future: Opportunities in Long-Acting Injectable Drug Development for Veterinary Care

**DOI:** 10.3390/pharmaceutics17050626

**Published:** 2025-05-08

**Authors:** HariPriya Koppisetti, Sadikalmahdi Abdella, Deepa D. Nakmode, Fatima Abid, Franklin Afinjuomo, Sangseo Kim, Yunmei Song, Sanjay Garg

**Affiliations:** Centre for Pharmaceutical Innovation, Clinical and Health Sciences, University of South Australia, Adelaide, SA 5000, Australia; haripriya.koppisetti@mymail.unisa.edu.au (H.K.); sadikalmahdi.abdella@mymail.unisa.edu.au (S.A.); deepa.nakmode@mymail.unisa.edu.au (D.D.N.); fatima.abid@mymail.unisa.edu.au (F.A.); olumide.afinjuomo@mymail.unisa.edu.au (F.A.); sangseo.kim@mymail.unisa.edu.au (S.K.); may.song@unisa.edu.au (Y.S.)

**Keywords:** long-acting injections, veterinary medicine, extended drug release, biocompatibility

## Abstract

Long-acting injectable (LAI) formulations have revolutionized veterinary pharmaceuticals by improving patient compliance, minimizing dosage frequency, and improving therapeutic efficacy. These formulations utilize advanced drug delivery technologies, including microspheres, liposomes, oil solutions/suspensions, in situ-forming gels, and implants to achieve extended drug release. Biodegradable polymers such as poly(lactic-*co*-glycolic acid) (PLGA), and polycaprolactone (PCL) have been approved by the USFDA and are widely employed in the development of various LAIs, offering controlled drug release and minimizing the side effects. Various classes of veterinary medicines, including non-steroidal anti-inflammatory drugs (NSAIDs), antibiotics, and reproductive hormones, have been successfully formulated as LAIs. Some remarkable LAI products, such as ProHeart^®^ (moxidectin), Excede^®^ (ceftiofur), and POSILACTM (recombinant bovine somatotropin), show clinical relevance and commercial success. This review provides comprehensive information on the formulation strategies currently being used and the emerging technologies in LAIs for veterinary purposes. Additionally, challenges in characterization, in vitro testing, in vitro in vivo correlation (IVIVC), and safety concerns regarding biocompatibility are discussed, along with the prospects for next-generation LAIs. Continued advancement in the field of LAI in veterinary medicine is essential for improving animal health.

## 1. Introduction

Humans and animals are physiologically distinct due to differences in absorption, distribution, metabolism, and excretion (ADME) patterns. Humans exhibit a fully functional drug metabolism with extensive cytochrome CYP450 enzymatic activity, moderate clearance, and a standard volume of distribution in adults [1]. However, animals are different in this case, as each species shows a different physiological mechanism [2]. Thus, “one-size-fits-all” does not work while designing dosage forms for different species. Understanding these differences is crucial for designing practical treatment approaches. In veterinary medicine, species-specific physiological differences influence the use and design of LAIs. Dogs have active CYP1A2 and 2B11 enzymes, leading to rapid drug metabolism and demanding sustained-release formulations. They have a larger volume of distribution and moderate injection tolerance [3]. However, cats, in this case, present more challenges to the use of LAIs due to their poor glucuronidation, sensitivity to excipients, and lower drug clearance and volume of distribution [4]. Livestock animals such as cattle, sheep, and pigs have efficient metabolism and high clearance, making them suitable for depot formulations. They tolerate injections well compared to companion animals [5].

Due to these physiological discrepancies, certain drugs are exclusively approved for use in animals and not in humans, and vice versa. For instance, Xylazine, used as a sedative, and as a muscle relaxant in horses and cattle, is not approved for humans as it can cause severe CNS depression, bradycardia, and respiratory depression, and its misuse is associated with mortality [6]. Similarly, drugs such as monensin are used as poultry feed and to control coccidiosis, but are known to be highly toxic even at low doses to humans. Ractopamine and carbadox, used to promote leanness in pigs and cattle and as an antibiotic for swine growth, respectively, are not approved for humans due to their severe cardiovascular effects and carcinogenic behavior in laboratory animals, raising concerns about safety [7]. To summarize, these variations are primarily due to species-specific physiological and pharmacokinetic differences, particularly in ADME. Understanding such distinctions is critical not only for optimizing pharmacological outcomes but also for meeting the rising demand for tailored veterinary therapeutics [8].

### 1.1. Clinical Need and Benefit

Companion animals such as dogs, fish, pigs, reptiles, cats, horses, rodents, and rabbits hold significant importance in people’s lives by offering company, responsibility, and affectionate responses [9,10]. Extensive evidence supports that having a companion animal is associated with animal-assisted therapy (AAT), which offers various health benefits. These include improved mood, enhancement of cognitive function, lower blood pressure, relaxation, and reduced symptoms of anxiety and depression [11,12,13,14]. This contributes to the rising demand for vaccines and pharmaceuticals for companion animals’ overall health and well-being. Additionally, there is an increasing demand from both companion animal owners and veterinarians for products that are easy to administer, promoting consistent use and ensuring owner compliance. Veterinary medicine plays an important role in ensuring the health and well-being of animals. However, the current routes of administration, such as oral and intravenous administration routes, often lead to a subsequent decline in drug levels below the therapeutic thresholds within a few hours, creating a “peak-and-valley” effect [15,16]. This subsequently results in the use of excessive drug concentrations that may induce severe adverse effects and harm, while insufficient levels render the treatment ineffective. Moreover, oral administration faces challenges in drug degradation before absorption due to the harsh conditions and enzymatic activities within the gastrointestinal tract, and first-pass metabolism further diminishes the bioavailability of numerous therapeutic compounds [17,18,19].

To overcome these challenges, the administration of multiple doses becomes necessary to sustain the drug levels within the therapeutic range. Due to the inherent limitations of conventional routes of drug administration, there is a clear demand for the development of advanced drug delivery systems that enable the continuous and sustained release of therapeutic agents [20]. These delivery systems are designed to keep the drug concentrations within the desired therapeutic window, reducing the risk of toxicity and side effects while enhancing the treatment effectively. Among such advanced drug delivery approaches, long-acting drug delivery systems have emerged as promising solutions with substantial potential in the pharmaceutical field [21].

Long-acting injectables (LAIs) offer numerous advantages including improved compliance, reduced dosage frequency, improved bioavailability, prolonged therapeutic activity, and better disease management. LAIs are especially beneficial for the management of chronic conditions such as cardiovascular, neurological, and psychiatric disorders. This approach enables the prolonged release of active pharmaceutical ingredients, maintaining safe and effective drug levels over an extended period. By minimizing the frequency of administration, LAIs offer economic benefits and overcome the challenges associated with non-adherence [22]. Similarly to their use in humans, LAIs offer additional advantages to veterinary medicine as they minimize the dosage frequency, promoting animal comfort and minimizing distress for caretakers [23]. LAIs have been a promising strategy in human pharmaceuticals for managing chronic conditions such as HIV and schizophrenia by reducing the dosing frequency and overcoming “pill fatigue” [24].

However, their development is challenged by complex in vivo behavior influenced by various formulation variables (that include particle size, lipophilicity, excipients), administration site physiology, and host response. Some of the outstanding products include Cabenuva^®^ (cabategravir/rilpivirin) for HIV [25], Invega Sustenna^®^ (Paliperidone palmitate), Risperdal Consta^®^ (risperidone) for schizophrenia [26,27], and Lupron Depot^®^ (leuprolide) and Depocyt^®^ (Cytarabine) for cancer [28,29]. Despite the benefits of LAIs, the risk of burst release, the potential for injection reactions, and the difficulty of depot removal in case of adverse events are always challenging. Additionally, trial-and-error formulation approaches, interpatient variability, and unclear drug release mechanisms remain barriers [30]. While the fundamental principles of LAIs are shared between human and animal products, such as sustained drug release and improved compliance, the veterinary field faces unique challenges. These include greater variability in physiology, differences in metabolism, diverse administration routes across species, and cost-effectiveness for large-scale livestock treatment, as discussed in earlier sections [20].

Conversely, opportunities in veterinary medicine include more flexibility with regulatory pathways, long dosing intervals due to less stringent patient monitoring, and the potential to control zoonotic disease transmission. Such contrasts underscore the need for species-specific formulation strategies and tailored delivery systems in LAIs for veterinary purposes. Animals, especially livestock and aggressive companion animals, often resist frequent dosing; LAIs can reduce repeated handling and improve treatment adherence, improving animals’ welfare [31]. Animals also vary in size from companion animals like cats to cattle and horses, making a universal design challenging. Large animals require high doses, which leads to challenges in injection volume, drug loading, and administration. Also, high-dose formulations should not impose drug load and should achieve therapeutic levels without exceeding the safe injection volume. LAIs with high doses can also increase viscosity, cause burst release, and compromise the injectability [21].

### 1.2. Long-Acting Injection for Local and Systemic Treatment

Long-acting delivery systems in veterinary medicine can be used for both local and systemic delivery, depending on the therapeutic need and target animal species. The choice of injection route significantly influences efficacy and patient compliance. Since LAIs are designed to provide a sustained drug release, their primary routes of administration are intramuscular (IM) and subcutaneous (SC) [32]. The choice of IM or SC is based on the drug properties, absorption rate, animal species, and therapeutic needs. The IM route offers a faster onset of action and absorption rate due to the high blood supply to the muscular tissue but can cause irritation and pain at the injection site. Unlike the IM route, the SC route offers a slow rate of absorption due to there being less vasculature in the subcutaneous skin layer but is less painful and better tolerated [33]. However, in the IM route, the formulation can be given in large volumes of 2–5 mL for small animals (such as dogs and cats) and 10–20 mL for large animals (such as cattle and horses), whereas the volumes are limited to 1–2 mL for small animals and 5–10 mL for large animals in the SC route [34,35].

In an experiment conducted by Texas Tech University (TTU), the management of pain and stress after the physical castration of piglets was measured by administering the drug in the IM and SC routes. No significant changes in their pain behavior were observed with both routes, but piglets that received drugs by the SC route showed a slight decline in feeding behavior, which was assumed to be due to local irritation or discomfort at the site of injection [36]. In another study by Tatiana et al., buprenorphine was given in the IV, IM, SC, and oral transmucosal route (OTM) in cats as a postoperative analgesic. It was observed that cat groups that received the IV and IM routes of buprenorphine exhibited better postoperative analgesia compared to the OTM or SC route [37]. These results indicate that the IM route is preferred when a faster onset of action and the burst release of drugs are desired, whereas for the slower uptake of drug molecules and controlled drug release for a longer duration from the first time point, the SC route is the preferred choice [38,39]. However, the literature suggests that LAIs formulated as implants, oil solutions, or suspended solutions are suitable for the IM route. Polymeric microparticles, in situ gels, hydrogels, and depot-forming systems are preferred for the SC route. Individual delivery systems are discussed in detail with respective case studies in the following sections.

Apart from systemic delivery, LAIs have also shown their significance in local drug delivery. In working animals such as horses, joint diseases are more prevalent, and traditional oral medications, systemic drug delivery, or topical delivery are not effective approaches. In such cases, local delivery via intra-articular injection is one of the beneficial approaches, as the direct administration of a formulation to the synovial joint holds the formulation supporting its extended drug release [40]. Although this area has its limitations, as there is a high diffusion rate of molecules, drugs injected in direct solution form may diffuse out quickly [41]. Thus, long-acting injections are being designed that can resist the synovial environment and deliver the drug for a longer duration [42]. LAI approaches, including polymeric materials, have been a successful approach for addressing local delivery in osteoarthritis. NSAIDs are the first line of drugs for this condition and often require multiple doses to reduce the pain in joints for horses. Scott et al. designed extended-release microspheres of flavopiridol that were injected via an intra-articular route into the horse’s joints. The formulation delivered the drug for up to six weeks and was determined as a viable alternative for other intra-articular medications available for equine joint diseases [43]. Similarly, a sustained release of Tacrolimus-loaded polymeric particles was developed, showing prolonged drug release for up to 42 days [44].

Recently, LAIs for intra-periodontal drug delivery have been a promising approach for animals, as periodontal diseases (PDs) are the most common bacterial infections caused by inadequate oral hygiene and poor diet contributing to bacterial build-up. The intra-periodontal route involves the direct delivery of drugs into the periodontal pocket (the space between the tooth and the gum tissue) where bacteria tend to accumulate [45]. Systemic drug delivery for treating periodontitis poses challenges in animals due to species-specific biodistribution and difficulty in repeated dosing. Studies report that around 80% of dogs aged three and older suffer from PD, which leads to serious health complications if left untreated properly [46]. In this context, LAIs present an efficient approach by enabling localized drug delivery, dose reduction, sustained release, and better treatment adherence for both animals and caregivers. In a study by Zhuang et al., hydrogels composed of minocycline HCl and calcium dextran sulfate administered into the pockets sub-gingivally in dogs provided an effective drug release for ten days with a significant reduction in inflammation compared to systemic therapy [47]. Similarly, PLGA-based nanoparticles loaded with minocycline have demonstrated a controlled release profile with in vivo studies [48].

## 2. Classification, Materials, and Design Approaches for Long-Acting Formulations

Parenteral administration methods, such as intramuscular, intravenous, or intradermal injections, bypass gastrointestinal degradation and first-pass metabolism. Long-acting injectables (LAIs) have been employed for numerous medical purposes, including eye disorders, pain control, nasal polyps, cancer treatment, birth control, and neurological conditions. This drug delivery strategy offers sustained release, making it ideal for managing chronic illnesses like HIV, age-related macular degeneration, and schizophrenia. Currently, available LAI products mainly focus on treating persistent conditions, such as mental health disorders, hormonal contraception, and cancer [49]. Various LAI types have been developed and utilized in clinical practice. These include prodrugs with linker systems, lipid and polymer vesicles, biodegradable polymer depot systems, nanoparticles, water-insoluble suspensions, protein PEGylation, and implants. Formulations like polymeric conjugates, nanoparticles, microparticles, hydrogels, liposomes, and microneedles have been developed to address existing challenges in cancer treatment [50]. LAIs are categorized based on their action mechanism, formulation characteristics, release kinetics, and therapeutic applications into (1) aqueous dispersions or solutions; (2) oily injections; (3) in situ depots; (4) microspheres; (5) implants; and (6) liposomes [51]. The subsequent sections discuss each formulation technology in detail. This article thoroughly overviews current clinical needs and potential applications for long-acting formulations in veterinary medicine, emphasizing their potential benefits in addressing veterinary practice challenges. It also explores advancements in drug delivery technologies and their potential impact on developing long-acting formulations for veterinary use.

## 3. Technological Approaches in Long-Acting Formulations

### 3.1. In Situ Depots

In situ depots have gained more attention as controlled drug delivery systems suitable for proteins and drugs for which oral delivery is challenging. The parenteral administration of depot systems increases bioavailability and reduces the dose and frequency of administration. In situ depots are usually designed as a drug delivery carrier solution administered subcutaneously or intramuscularly, which forms a depot at the site of administration and slowly releases the drug [52]. Despite their vast benefits and scientific outcomes, it is still unclear whether prolonged drug release for a few hours or sustained release for days by depot systems is to be considered long-acting. However, an ideal depot system can be defined as a formulation that can deliver the drug at a controlled and determined rate within the therapeutic range for the desired time [53]. Drug interactions with the delivery system should be considered a primary factor in ensuring stability and preventing the leakage of drugs from the matrix [54]. Depots generally could be oil-based, lipid-based, or polymer-based, with each having significance due to its chemical structure, biodegradability, biocompatibility, and solubility [55]. Some of the key advantages and disadvantages of in situ depots developed using different materials are described in Table 1.

In situ-forming systems are widely approved and commercially available delivery systems for humans and animals. Atrigel^®^ is one of the first in situ depot systems approved as a long-acting drug delivery technology. It has been widely used for humans in the delivery of drugs such as doxycycline for tissue regeneration, and leuprolide acetate for advanced prostate cancer. It is known for its sustained release of drugs for a duration of 1 to 6 months [67]. This technology is now being used for the delivery of antigens, such as the inactivated pseudorabies virus to pigs, which results in increased serum antibody levels for more than 90 days with a single shot [68]. Another popular depot technology in the veterinary industry is SABER^TM^ (Sucrose Acetate Isobutyrate Extended Release), which is used for the delivery of peptides, proteins, and drugs over days to months for both animals and humans. Sucrose acetate isobutyrate is a combination of esterified sucrose derivatives with six isobutyrates and two acetates. This form of sucrose exists as a hydrophobic liquid and does not crystallize. It is approved as a food additive and pharmaceutical excipient in over 40 countries [69]. The commercially approved SABER-carrying progesterone has shown a sustained release of up to 28 days in mares [70,71]. POSIDUR™ SABER^®^, a bupivacaine formulation, is currently in a Phase III clinical trial for postoperative pain management [72]. Sekar et al. have conducted studies on the subcutaneous delivery of recombinant human growth hormone (rhGH) by SABER for systemic and local delivery. It has shown successfully elevated insulin-like growth factor (IGF-1) levels in monkeys and rats for a period of weeks to 1 month. The same group has conducted studies in beagles for the treatment of osteoarthritis by administering rhGH via the intra-articular route. It has shown a sustained drug plasma concentration for three months [73]. Similarly, ALAZAMER^®^, OncoGel^TM^, and ReGel^®^ systems are other types of clinically approved depot systems [74,75,76].

In another study by Geng et al., in situ gels were developed to minimize the dosing frequency of florfenicol for bacterial infections in pigs. The study was conducted on twelve pigs, with six in each group. One group was given a conventional injection of florfenicol, and the other was given florfenicol in situ-forming gel by the IM route. They achieved successful results in the prolonged release of florfenicol with an increase in its elimination half-life, thereby reducing the frequency of dosing [77]. Buprenorphine is frequently used as an analgesic in pain management for laboratory animals. However, the major drawback is the short half-life of the drug, which necessitates multiple administrations. To solve this, Viktoria et al. have developed a PLGA-based microparticle formulation with buprenorphine and administered it to laboratory mice. The formulation after administration by the IM route formed a depot at the site of injection that showed a burst release, i.e., 30% drug release within one hour, followed by sustained release of the drug for seven days. This was an effective approach to the management of pain with a reduced frequency of administration [78]. Similarly, Audrey et al. have investigated celecoxib in polymeric in situ-forming gels for the treatment of osteoarthritis in horses. A progressive loss of cartilage in the joint and the disruption of its integrity are the most common causes of pain in horses with osteoarthritis. Non-steroidal anti-inflammatory (NSAIDs) drugs are used as the first line of drugs for treatment but are known to show systemic side effects upon local drug delivery. To minimize the drug disposition to organs and retain the drug on the target tissue, an intra-articular administration of celecoxib (COX-2 inhibitor) was loaded into in situ gel-forming systems. After the administration of these systems, high drug concentrations were found in the target joint for a prolonged duration, limiting systemic exposure. Histological examinations of the tissue have shown a good tolerance for high doses of celecoxib without any damage to cartilage, suggesting that an in situ gel-forming system is an effective approach [79].

### 3.2. Hydrogels

Hydrogels are three-dimensional polymeric networks capable of absorbing significant quantities of aqueous solutions, including water and physiological fluids, while retaining their structural stability and insoluble nature [80,81,82]. In recent times, hydrogels have emerged as a pivotal entity in the realm of drug delivery, assuming a profound role as drug carriers and exhibiting potential as a platform for accomplishing sustained and prolonged drug release [83]. In addition to the fact that hydrogels possess the ability to mimic the extracellular matrix (ECM), they create an optimal milieu for cellular proliferation and tissue regeneration, which has resulted in considerable interest in their use in regenerative medicine [84,85]. With their exceptional ability to swell, their high permeability, and their biodegradability, they are extensively suitable as vehicles for drug delivery [86,87]. Depending on the fabrication method, they are classified into physically crosslinked and chemically crosslinked hydrogels as shown in Figure 1 [88].

Physical crosslinking of hydrogels involves the utilization of weak forces, such as ionic interactions, crystallization, stereo-complex formation, freeze-thawing, hydrogen bonding, and hydrophobic interactions, to establish connections between water-soluble polymer chains. In contrast, using the crosslinkers to bond polymers leads to the formation of chemical hydrogels [89]. Various approaches and methodologies have been reported for the fabrication of hydrogels through crosslinking, including chemical crosslinking, grafting, Michael addition, click chemistry, Schiff base reaction, enzyme-mediated reactions, photocrosslinking, radical polymerization, condensation reactions, enzymatic reactions, and high-energy radiation [90]. Physically crosslinked hydrogels are reversible gels that are easier to manufacture as they do not need a chemical crosslinker. Chemically crosslinked hydrogels are stable due to the covalent bonds between polymer chains, allowing for the tuning of parameters such as internal network pore size, gelation time, and degradation time [91,92].

Hydrogels exhibit diverse classifications based on their physical state (solid, semi-solid, and liquid), crystallinity (non-crystalline, semi-crystalline, or crystalline), electrical charge (anionic, cationic, amphoteric, or non-ionic), and degradation pattern (biodegradable or non-biodegradable) [93]. Injectable hydrogels possess excellent physicochemical properties, particularly viscosity, which allows them to be injected in situ, especially for non-invasive surgery. In addition, injectable products can trigger inflammatory reactions such as immune-mediated reactions, which demand that the product be biocompatible and non-toxic. Hydrogels are biocompatible, non-toxic, and possess high porosity, which makes them suitable for a higher drug loading capacity, facilitates the better movement of nutrients, and improves adaptation to their surroundings. One of the challenges of using hydrogels for drug delivery is their hydrophilic nature, which poses difficulties while loading and delivering hydrophobic drugs. Additionally, the relatively low tensile strength of hydrogels may lead to the early release of drugs [86].

Natalia et al. have designed a chitosan-based hydrogel for the treatment of bacterial infections in cows. The in vivo study demonstrated sustained antibacterial activity, with the therapeutic concentration maintained locally for up to 72 h, indicating prolonged drug residence time and enhanced local bioavailability [94]. Similarly, polyvinyl alcohol-based hydrogels were formulated for the controlled release of tylosin tartrate and oxytetracycline, commonly used veterinary antibiotics. When tested in rats alongside standard oral formulations, the hydrogel system exhibited significantly prolonged drug release, sustaining the plasma concentrations for up to 120 h. This suggested a marked extension in the drug’s half-life and reduced fluctuations in plasma concentration levels, minimizing peak-trough effects and enhancing pharmacokinetic stability [95].

Further advancement in hydrogel design has led to a stimuli-responsive system. Fatma et al. engineered a hybrid in situ pH-sensitive hydrogel composed of curcumin-loaded niosomes and doxycycline-loaded chitosan–sodium alginate nanoparticles. This multifunctional delivery system was tested in Brucella melitensis biovar 2-infected guinea pigs. Pharmacokinetic analysis showed sustained systemic drug exposure and an extended mean residence time (MRT) in comparison to conventional formulations. The hybrid hydrogel not only improved bioavailability but also facilitated controlled and site-specific release, highlighting the therapeutic potential of the intelligent hydrogel system for veterinary applications [96]. Similarly, hydrogels can also be used with natural products such as hyaluronic acid as a biomaterial, which can be used for tissue regeneration [97]. Xiaodan et al. developed a curcumin-loaded hydrogel using gelatin methacryloyl (GelMA) as a sustained anti-inflammatory platform to promote cartilage regeneration in immunocompetent animals. The hydrogel system achieved a controlled release of curcumin over six weeks, indicating prolonged drug retention at the target site. In vivo evaluations demonstrated consistent local bioavailability of curcumin, contributing to reduced inflammatory markers and improved cartilage repair. The extended drug release suggested the prolonged half-life and increased mean residence time (MRT) of curcumin in the joint microenvironment. This delivery system enhanced the therapeutic window while minimizing the need for repeated dosing, highlighting its suitability for chronic inflammatory conditions in veterinary orthopedics [98]. In conclusion, hydrogels represent a transformative advancement in drug delivery systems, particularly for applications in sustained release.

### 3.3. Implants

Pre-fabricated solid implants are made from biodegradable or non-biodegradable polymers, generally cylindrical in shape, with a 10–35 mm length and 1 to 3 mm diameter [99]. They offer the local as well as the systemic delivery of drugs. Non-biodegradable implants necessitate surgical removal after the completion of the treatment, whereas biodegradable systems include polymers that degrade in the body over time at the site of application after drug release [100]. Despite the drawback of requiring surgical removal, pre-fabricated implants offer numerous benefits. They enable site-specific drug delivery for extended durations, ranging from months to years while being cost-effective and accommodating a wide range of sizes and doses. These implants minimize systemic side effects by delivering drugs locally [101]. Unlike other parenteral delivery systems, which often pose sterilization challenges, pre-fabricated implants can be easily sterilized using steam autoclaving or irradiation. Additionally, the option for surgical retrieval in case of adverse reactions enhances their safety profile [102].

Biodegradable implants are generally preferred over non-biodegradable ones because they naturally degrade, erode, or are excreted over time, improving both patient acceptance and compliance [103]. Compared to other injectable delivery systems, implantable drug delivery systems provide the flexibility of retrieval upon the observation of any side effects [104]. Based on their biocompatibility and good mechanical strength, biodegradable polymers such as polylactic acid (PLA), poly(glycolic acid), poly(lactic-*co*-glycolic) acid (PLGA), and polycaprolactone (PCL) are commonly used [21]. Implants can be prepared using various technologies such as compression, solvent casting, hot melt extrusion, injection molding, electrospinning, and 3D printing [104]. Compression is the most preferred method and is used in various marketed implant products such as FINAPLIX^®^, SYNOVEX^®^, RALGRO^®^, and REVALOR^®^ [105].

The commercial formulation REVALOR XS is an extended-release implant with one dose containing 200 mg of trenbolone acetate and 40 mg of estradiol, comprising four uncoated pellets (for initial drug release up to 80 days) and six coated pellets with a biodegradable polymer which provides drug release up to 200 days. Each pellet contains 20 mg trenbolone acetate and 4 mg estradiol, and they are indicated for steers fed in confinement [106]. SUPRELORIN is a sustained-release implant that includes deslorelin acetate in a matrix of hydrogenated palm oil and lecithin given subcutaneously. It is used temporarily to induce fertility in healthy, sexually mature, non-neutered ferrets and dogs. Infertility is achieved from 6 weeks up to at least 6 months after initial treatment [107]. The non-biodegradable polymers include silicones and poly(ethylene-vinyl acetate). COMPUDOSE is a silicone implant given subcutaneously containing 25.7 mg estradiol; it provides a controlled release for 200 days. It is also coated with no less than 0.5 mg of oxytetracycline powder as a local antibacterial to prevent infection at the injection site. The release of estradiol from the matrix system follows a diffusion mechanism [108].

The release of drugs from implants follows different mechanisms, which have been categorized into four pathways: passive diffusion, matrix degradation, controlled swelling, and osmotic pumping. Passive diffusion or permeation is driven by a concentration gradient where the drug molecules move from one region to another until equilibrium is achieved. Some of the commercial products that deliver drugs by this mechanism are Compudose, Synovex, and Ralgo [109]. In matrix degradation, drug release can occur after the polymer degrades or through a combination of processes such as hydrolysis of the ester bond, enzymatic degradation, or polymer swelling. These processes depend on the polymer characteristics and molecular weight as seen in the marketed products Spanbolet and monensin RRD [110]. In controlled swelling, the release rate is regulated by the rate of the solvent penetration matrix [111]. In osmotic pumping, the osmotic pressures act as driving forces for the transport of drugs from the system. Commercial products such as VITS and ALZET adapt osmotic pumping as a drug release mechanism [21].

### 3.4. Oily Solutions

The development of simple oil-based systems, including oil solutions and oil suspensions, started in the 1950s, and its products have dominated the LAI market [112]. Many lipophilic drugs have been formulated into oil-based systems as they can overcome several of the challenges associated with aqueous formulations, including poor drug solubility, instability, and potential irritation [105]. They are generally easy to prepare, produce less irritation, have good long-term stability, and possess higher safety than other injectables [113].

To formulate oily solutions, vegetable oils, including castor oil, sesame oil, safflower oil, soybean oil, peanut oil, cottonseed oil, fractionated coconut oil, and medium-chain triglyceride oils with high biocompatibility, have commonly been used [114]. Oil selection is one of the key factors that affect drug release kinetics. The partition coefficients of the drug between the oily solvent and the surrounding tissues at the injection site directly impact the rate of drug absorption [114]. In addition, a strong correlation has been demonstrated in several in vivo animal studies between the viscosity of the oil and drug absorption, where a higher viscosity of the injectable oil results in an increased plasma half-life [115,116,117]. Practically, oil-based LAI products are often limited to a shorter retention time of up to 4 to 6 weeks and commonly require repeated dosing every 2 to 4 weeks [113].

In oily suspensions, achieving uniformity upon resuspension is essential to maintaining consistency in the drug content, which is often noticeable when only oils are used in manufacturing. Thus, Foster and Kiefer patented the addition of small volumes of water to the oily suspension to improve its dispersibility. The inclusion of small amounts of water directly or from the API or its excipients will aid in the dispersion of the drug and the control of flocculation. Reports showed that this approach was successful in maintaining uniform dispersion within 10 s of shaking even after 6 months of storage for a ceftiofur HCL suspension in cottonseed oil, achieving 0.39% compared to an equivalent suspension with 0.20% water, which was not adequately re-dispersed after 6 months of storage [118]. Various marketed formulations of oil solutions and suspensions are available and listed in Table 2.

In a pharmacokinetic study of Boostin^®^-S, 500 mg of rbST was administered at days 0, 14, and 28, which resulted in the peak plasma bST level at 24 h (28.0 ± 6.56 ng/mL) after administration, with its plasma half-life determined between 100 and 137 h [138]. When 1.5 mL of Depodine was given to *Farahani ewes*, a sheep variety, three weeks before mating and a repeated dose given nine weeks after, the serum inorganic iodine concentration increased from 2.39 to 11.2 µg/dL during the 180-day study period [139]. Bimoxyl LA achieved its peak concentration in bovine plasma at 3.5 µg/mL within one hour of administration [140]. Excenel^®^ RTU was given intramuscularly or subcutaneously at 3–5 mg/kg body weight every 24 h for 3 consecutive days. It was administered intramuscularly at 5 mg/kg to healthy pigs, and its maximum plasma concentration was found to reach 36.1 ± 6.2 µg/mL within 1.26 ± 0.21 h of the administration [141]. As opposed to other oily suspensions in this category, the release of vitamin B12 from SMARTShot^®^ B12 is mainly dependent on the erosion of the PLGA polymer, creating pores for vitamin B12 to diffuse throughout the 4 to 6-month period [105]. A single subcutaneous injection (1 mL) of SMARTShot^®^ B12 given to lambs significantly increased the serum vitamin B12 concentration from 508 ± 21 pmol/L to its peak concentration of 1950 pmol/L within 31 days, before it returned to 577 pmol/L within 246 days [142].

### 3.5. Suspended Solids

Suspension-based long-acting injectables are systems that are suspended in suitable aqueous (water) or non-aqueous (vegetable and synthetic oils) vehicles based on the physio-chemical properties of the drug [21]. The formulation of the drug into a suspension form improves its stability and provides controlled drug release for several weeks to months after a single injection. This approach benefits chronic conditions by improving the feasibility of administration in a veterinary clinic [143]. Compared to other delivery systems, the advantages of the suspension approach include the improved stability of the drug and formulation, higher drug loading, cost-effectiveness, and ease of scale-up [144]. Suspensions are administered via the intramuscular or subcutaneous route, and a depot is formed at the site of injection from which drugs are slowly absorbed, providing a prolonged drug concentration in the body. Suspension systems are categorized as nanocrystals when the drug exists in the crystalline form. The nanocrystal absorption pathway involves absorption by blood capillaries, allowing for a direct entry into the systemic circulation. Further, some of them are drained into thoracic lymphatic vessels after absorption, eventually reaching the systemic circulation [118]. Finally, nanocrystals are absorbed by macrophages in the lymphatic system to form a secondary depot. Certain hydrophobic drugs, before the conversion into nanocrystals, are modified into prodrugs and are then formulated as a nanosuspension [145]. In chronic conditions such as diabetes, when a suspension of insulin is injected subcutaneously, it has shown a longer duration of control of blood glucose in humans and animals [118]. Thus, this approach has been widely adopted for developing insulin formulations. In further studies by Greco et al., amorphous or crystalline suspensions of zinc and protamine insulins were developed for prolonged release. Their studies concluded that glycemic control can be achieved by selecting the appropriate insulin type based on the duration of action and dosage regimen [146].

The commercial formulation PEN BP-48 is a subcutaneous aqueous suspension containing penicillin G procaine and penicillin G benzathine salts, with sodium carboxymethylcellulose as the aqueous vehicle. It is used in the treatment of bacterial pneumonia (shipping fever), and upper respiratory infections such as rhinitis or pharyngitis in cattle and beef [21]. Buckwalter and co-workers evaluated the effect of the vehicle and particle size on the absorption of procaine penicillin G after intramuscular administration. They found that the drug’s pharmacokinetic profile was affected by its particle size, salt form, and the nature of the vehicle used. Their results indicate that smaller particles under 5 µm in size within an oil–aluminum stearate vehicle are more effective than large particles in delaying the absorption of procaine penicillin G [147]. Pfizer’s long-acting lyophilized product CONVENIA^®^ is a reconstitute suspension of cephalosporin cefovecin sodium used for cats and dogs for bacterial infections. The antimicrobial effect lasts up to 14 days after a single injection [148]. PROGRAM^®^6, a product by Novartis, is a prepacked, ready-to-use syringe containing a sterile suspension of 10% lufenuron which is effective for six months in cats for flea control [105].

### 3.6. Liposomes

Veterinary pharmaceutical products require safety, cost-effectiveness, ease of use, and low toxicity. Advances in nanotechnology have enabled drug carriers that improve bioavailability, reduce toxicity, and evade the immune system, allowing for prolonged release [149]. Among these, liposome-based drug delivery has gained significant interest in both human and veterinary applications [150]. Discovered in the 1960s, liposomes are spherical, bi-layered lipid structures mimicking the cell membrane [151,152]. They offer high biocompatibility, biodegradability, targeted delivery, reduced systemic toxicity, and enhanced pharmacokinetics, making them effective drug-delivery vehicles [153]. They are classified based on composition, phospholipid layers, and method of preparation into small unilamellar vesicles (size ranging from 20 to 100 nm), large unilamellar vesicles (size > 100 nm), giant unilamellar vesicles (size > 1000 nm), multilamellar vesicles (size > 500 nm), oligolamellar vesicles (size ranging from 100 to 1000 nm), and multivesicular liposomes (size > 1000 nm) [154]. Their surface can be modified into stealth liposomes for long circulation or as immunoliposomes for targeted delivery [155]. Ideal liposomes are known to be 50 to 200 nm, which increases the residence time of liposomes in blood circulation and enhances the duration of drug release [156].

Liposomes made of phospholipids and cholesterol with sizes from 0.02 to 5 µm are capable of prolonged release, making them suitable for depot formulations. Surface charge, influenced by lipids, affects drug release, biodistribution, and clearance [157]. Stability is improved by longer phospholipid chains, cholesterol addition, and the use of negatively charged phospholipids with magnesium or calcium to prevent aggregation [158,159,160]. Traditional preparation methods include thin-film hydration, reverse-phase evaporation, ethanol/ether injection, and detergent depletion, each with its advantages and limitations [161,162,163,164,165]. Recent techniques, such as microfluidics, freeze-drying, and supercritical fluid extraction, enhance scalability and produce a controlled particle size but have issues like aggregation and are more resource-intensive [166,167]. Liposomes can deliver hydrophilic, hydrophobic, or amphiphilic drugs, making them versatile for various applications [168,169,170].

Anti-cancer drugs often face poor bioavailability and high toxicity. Liposomes enhance the delivery by encapsulating hydrophobic drugs and exploiting the tumor’s leaky vasculature for targeted release via enhanced permeation and retention (EPR) effects [171,172]. The first clinical trials of this delivery technology for veterinary applications began in dogs for canine splenic hemangiosarcoma (HAS) by administering liposome-encapsulated muramyl tripeptide, resulting in positive results and the enhanced efficacy of muramyl peptide in comparison to the conventional IV administration of free peptide [173]. Since then, liposome technology for animals has gained attention for its application to treat various other diseases. In a pilot study conducted by Hauck et al., low-temperature-sensitive doxorubicin-loaded liposomes were administered to 21 dogs with sarcomas. After administration into the tumor site, 100% of the drug was released within 20 s, triggered by the tumor temperature. Out of the 21 dogs, 12 had a tumor volume reduction to <50% and 6 had a partial response in terms of tumor volume. This technology paved the way for a novel approach to delivery into tumors [174].

Similarly, in recent years, vaccines have evolved to use synthetic peptides and recombinant proteins to avoid the risks of live-inactivated pathogens. These subunits require adjuvants for better immunity. Modified liposomes, such as cationic types, enhance uptake by antigen-presenting cells, boosting T-cell responses [175]. Wenzhe et al. investigated the use of liposomal delivery for fimbriae antigens (SEF14 and SEF21) targeting *Salmonella enterica serovar Enteritidis* (*S. Enteritidis*) in chickens. The study demonstrated a significant reduction in intestinal bacteria and increased IgA and IgG responses, indicating that liposomal delivery effectively induced systemic and mucosal immunity to suppress *S. Enteritidis* infection [176]. This delivery method has also been applied to vaccine development for agricultural animals. Ovine abortion, a common issue in sheep caused by the parasite *Toxoplasma gondii*, is currently managed with Toxovax, a live vaccine. As a safer alternative, a subunit microneme protein MIC3 was designed to bind host cells. To efficiently deliver this protein, a liposomal DNA vaccine carrying the plasmid encoding the mature MIC3 subunit was developed. This formulation was tested in 36 ewes and showed a robust immune response against the parasite, supporting the further exploration of liposome-based vaccines as a safer and more effective option for livestock [177].

Effective pain management in animals, especially in pets, farm animals, and laboratory animals, is critical and challenging due to the shorter half-life of analgesics. Liposomes have been proven to be an excellent option to address these issues. Depofoam bupivacaine was the first single-dose liposomal injection tested successfully in rabbits and dogs. Similarly, liposomal opioid formulations with oxymorphone and hydromorphone released drugs over 5 days in rats, preventing hyperalgesia [178]. In horses with induced osteoarthritis, diclofenac liposomal cream significantly reduced lameness compared to phenylbutazone and is now approved by the United States Food and Drugs Administration (USFDA) for pain management in horses [179]. This delivery technology also holds promise for immunotherapy and gene delivery in veterinary medicine. For example, liposomal clodronate, initially developed to prevent osteoarthritis, was found to be effective in suppressing tumor-associated macrophages and including apoptosis in canine hemangiosarcoma. In another study, liposomal interleukin-2 (IL-2) demonstrated antitumor immune responses in dogs with pulmonary metastasis, showing better tolerance and fewer side effects than free IL-2 [180]. These findings highlight liposomes as a versatile and safer platform for delivering drugs and vaccines in animals. Some of the clinical and preclinical trials of liposomes reported in the literature are mentioned in Table 3.

### 3.7. Microparticles/Microspheres

Microspheres are small spherical-shaped particles with a particle size range of 0.1 to 200 µm and are made up of polymers that can be biodegradable or non-biodegradable. They can encapsulate a wide range of drugs, including hydrophilic and hydrophobic drugs, nucleic acids, or proteins [192,193]. Polymeric microspheres offer significant chemical adaptability and enable controlled, sustained drug release. They can be customized to suit the specific needs of different animal species, addressing the unique challenges in veterinary medicine. This delivery method decreases the frequency of handling and dosing, reducing animal stress while enhancing therapeutic outcomes through prolonged drug release. Additionally, the use of biodegradable polymers removes the necessity for surgical removal, thereby lowering the associated risks and costs [194]. Biodegradable polymers such as poly(lactic-co-glycolic) acid (PLGA), polycaprolactone (PCL), and polylactic acid (PLA) are commonly used to formulate microspheres due to their biocompatibility and controlled degradation profile. The critical factors that influence microspheres are the fabrication technique, type of polymer, and copolymer, which influence their properties and release kinetics [195]. The use of biodegradable and biocompatible polymers reduces the toxicity and enhances the bioavailability of drugs. Drug release from polymeric microspheres occurs via diffusion, dissolution, surface erosion, and bulk erosion. In diffusion, the dissolution fluid penetrates the shell, creating pores for drug diffusion. In dissolution, the polymer coat dissolves, with the release rate depending on its thickness and solubility [196]. Erosion involves bulk degradation, where hydrolysis causes swelling and gradual drug release, and surface erosion, driven by enzymatic hydrolysis or oxidation, often influenced by inflammatory responses. Enzymes degrade the polymer at the interface, releasing the drug. Factors like microsphere size, polymer molecular weight, copolymer composition, drug-polymer interactions, and glass transition temperature significantly influence the degradation and release rate [197].

PLGA is a copolymer of lactic acid (LA) and glycolic acid (GA); LA contributes to the hydrophobic component, and GA to the hydrophilic nature of the PLGA. The amount of LA influences the degradation kinetics of the PLGA; the higher the LA content, the greater the increase in the hydrophobicity of the polymer, and the slower the drug release and polymer degradation [198]. Microspheres composed of PLGA release the drug by three synergistic mechanisms: Initial burst release occurs by diffusion or surface erosion, followed by dissolution and swelling, causing the erosion of the drug from the pores or the matrix of PLGA. Finally, the hydrolytic degradation of PLGA to LA and GA releases the drug and metabolizes the polymer [199]. Similarly, PLA-based formulations show a sustained release profile through a combination of diffusion and erosion. PLA forms a viscous layer upon degradation that shifts the release dynamics to a combination mechanism. Also, under hydrolytic degradation, it forms LA that contributes to further degradation based on its molecular weight [200]. The pharmacokinetic evaluation of a novel compound, orntide acetate, from PLGA and PLA shows its prolonged drug release for extended periods from weeks to months, depending on the molecular weight and copolymer composition [201]. PCL is also a hydrophobic, semi-crystalline aliphatic polymer widely used for LAIs, particularly for drug release over months to years [202]. Polymeric particles composed of PLC release the drug primarily by surface erosion and diffusion followed by degradation and erosion-controlled release. The degradation of PLA is much slower than PLGA and PLA, as the hydrolytic degradation of ester bonds is slow. Erosion plays a key role in drug release at the later stages, typically from six months to years [203].

Companion animals undergoing chemotherapy often face severe side effects from anti-cancer drugs, which are a common cause of their mortality. To address this, Gavini et al. formulated microspheres using PLGA to encapsulate carboplatin, a widely used chemotherapeutic agent. In preliminary in vivo studies on rats, the subcutaneous administration of these carboplatin-encapsulated microspheres achieved 90% sustained drug release over 21 days. There were no toxic or local reactions from the drug-free microparticles, indicating a safe and effective delivery system [204]. Similarly, Yang et al. formulated tilmicosin-loaded gelatin microspheres to treat bacterial respiratory diseases in livestock. This approach extended the drug’s elimination half-life, providing a sustained release for up to 9 days with maximum distribution to the lungs, reducing the need for frequent dosing [205]. Other studies have explored cephapirin-loaded PLGA microspheres and meloxicam–PLGA microspheres for sustained delivery in cattle [206,207]. In conclusion, the development of polymeric microsphere-based drug delivery systems offers a promising solution to enhance therapeutic efficacy and minimize side effects in both companion animals and livestock. These systems provide sustained drug release, reduce the need for frequent dosing, and demonstrate safety and effectiveness, making them valuable tools for improving treatment outcomes in veterinary medicine.

### 3.8. Nanoparticle-Based Approaches

Nanoparticles are classified into lipid-based, polymer-based, and metal-based delivery systems based on their primary material of use for the matrix. Polymer-based delivery systems include dendrimers, nanospheres, niosomes, and polymeric nanoparticles, whereas lipid-based delivery systems include nanoliposomes, solid lipid nanoparticles (SLNs), and lipid vesicles, while metal-based delivery systems include nanotubes, gold nanoshells, and metal colloids [208]. In comparison to all of the nanocarriers, polymeric nanoparticles and liposomes are mostly explored for veterinary drug delivery. Polymeric nanoparticles are colloidal particles with sizes ranging from 1 to 100 nm, usually prepared using biodegradable or non-biodegradable polymers that stabilize the drug with or without surfactants. Non-biodegradable polymeric nanoparticles have shown chronic toxicity and higher immunological responses over long-term use. Consequently, biodegradable polymers have become the preferred choice for nanoparticles [209].

Based on the method of preparation and the drug’s solubility, the active ingredients are either dissolved, embedded, or encapsulated in the matrix of the nanoparticles. Generally, polymeric nanoparticles are nanocapsules or nanospheres, depending on the method of preparation. In the nano capsule, the drug is encapsulated by the layer of polymer, whereas the drug is dispersed in the matrix of the polymer in nanospheres [210]. Polymeric nanoparticles provide the advantages of the modification of drug release, particle size, and zeta potential by altering the polymer chain length and surfactant concentration [211]. Although various types of polymers are available, PLGA is a USFDA-approved and commonly used polymer. The drug release profile of nanoparticles prepared using PLGA can be altered by adjusting the PLA-to-PGA ratio [212]. In recent years, PLGA nanocarriers have been applied in veterinary medicine for drug delivery to the central nervous system, such as delivering temozolomide to treat brain tumors in dogs [213]. Paccal Vet-CA1 is a sterile lyophilized powder of paclitaxel, composed of polymeric nanoparticles in the size range of 20 to 40 nm, and is given intravenously to dogs [214]. The Imrestor injection includes a 15 mg dose of PEGylated bovine granulocyte colony-stimulating factor for treating inflammation in the breast tissues of cows [215].

A study conducted by Feldhaeusser et al. evaluated the efficiency of polymeric nanoparticles containing Platin-M nanoparticles (modified Pt (IV)-prodrug of cisplatin) against the canine glioma J3TBG and SDT3G glioblastoma cell lines. The in vitro cell viability experiments showed that nanoparticle formulation was 200 times greater than the carboplatin SDT3G glioblastoma and 130 times greater in the glioma J3TBF cell line after 72 h of incubation. The in vivo safety and biodistribution studies in female beagles were performed by administering a single intravenous injection. The results demonstrated a higher accumulation of Platin from T-platin-M nanoparticles in the brain in comparison to the other analyzed organs [216]. SLNs are stable colloidal systems prepared using solid lipids, which encapsulate the drug in solution or suspended form in the solid lipid core stabilized with the surfactant. In comparison to liposomes, SLNs are more stable physically. Studies have shown that SLNs can be used for providing sustained drug release. Han et al. prepared solid lipid nanoparticles consisting of tilmicosin by oil in water emulsion–solvent evaporation for the treatment of mastitis. SLNs were prepared using hydrogenated castor oil as a lipid core and showed a sustained serum level of tilmicosin for up to 8 days in comparison to the same dose of the drug given alone in a phosphate buffer [217].

## 4. Characterization Methodologies for Long-Acting Formulations

### 4.1. Morphological Examination

Any drug delivery system is analyzed for its surface and particle size using scanning electron microscopy (SEM) and a zeta sizer. SEM is a high-resolution technique widely used to evaluate the surface of implants, microspheres, nanoparticles, and liposomes [218]. Sample preparation involves spreading the sample on the aluminum sample mount and sputter coating with gold or palladium to enable the conductivity of the surface. The internal structure of the hydrogels can also be examined using the cryo-SEM in which the sample in liquid state can be analyzed [219]. The particle size distribution (PDI—polydispersity index) and zeta potential can be determined by the zeta sizer for examining the stability of the nanocarriers and suspended solids. The particle size of suspensions can also be studied using photon correlation spectroscopy or Coulter counter after suspending the formulation in water or electrolyte solution [220].

### 4.2. Rheological Properties

The rheology of LAIs is a critical factor in their development and performance. Rheological properties such as viscosity, elasticity, and flow behavior play a key role in determining the injectability, stability, and drug release kinetics, and are measured using a rheometer. Viscosity is a fundamental rheological property that influences the flow behavior and injectability of LAIs. It is defined as the measure of a fluid’s resistance to flow under applied stress. The high viscosity of the formulations poses challenges in drug delivery. Evaluating the viscosity of in situ-forming gels or implants is essential, as it influences the drug release. Both pre- and post-thermo-conversion should be measured for the implants, with injectability largely being affected by the viscosity before injection [221]. Many of the LAIs exhibit a shear-thinning behavior, where the viscosity decreases under increasing shear rate; this allows the formulation to easily flow through a needle during administration, maintaining its structure at rest. For example, organogels and hyaluronic acid-based systems have shown shear-thinning behavior, enabling their successful injection via autoinjectors [222].

### 4.3. Differential Scanning Calorimetry (DSC)

DSC can be used to examine the physical state of an API in the formulation. The physical mixture of the drug and excipient is heated at a controlled rate, and any changes in the state are observed. The DSC thermogram of the formulation is then compared with the physical mixture and the individual components to understand their behavior in the formulation. This method has also been widely used in studying nanocarriers and aggregates [223,224]. For example, a study conducted by Xing et al. revealed the absence of the drug’s melting point in thermograms, indicating the successful encapsulation of the drug in the polymer matrix in the amorphous state. Changing the crystalline state to amorphous is beneficial and enhances the controlled release profile, which is often desired in veterinary applications [225]. Furthermore, DSC analysis is used to detect the glass transition temperature (Tg) of the polymers, providing insights into the thermal stability of the formulation. Understanding Tg is essential for predicting how the formulation will behave under storage and physiological conditions, ensuring the efficacy and safety of drug delivery dosage forms [226].

### 4.4. Thermo-Gravimetric Analysis (TGA)

The thermal behavior of the material can be studied using TGA. This technique is used to measure the change in the mass of a sample as it is heated or cooled over time. It is used to analyze the thermal stability and composition of materials. It can also be used to determine the degradation temperatures in organic solvent-based formulations [227]. It is a vital technique used to assess the thermal stability of LAIs. In veterinary medicine, particularly for formulations with biodegradable polymers, it helps in determining the decomposition temperature of both active ingredients and polymers. In a study conducted to evaluate the thermal behavior of PLGA-based microspheres, TGA thermograms revealed distinct weight loss steps corresponding to the degradation of the polymer and encapsulated drug indicating the formulation’s stability only up to a particular temperature [228]. Such understandings are essential in confirming the accurate temperature at which a formulation can be stable without compromising on its safety and efficacy under thermal stress conditions [228].

### 4.5. Fourier-Transform Infrared (FTIR) Spectroscopy

FTIR helps in the investigation of any possible interaction between drugs and excipients, where each sample is scanned at a range of different wavelengths [223]. In veterinary formulations, confirming the integrity of functional groups post-formulation is critical to confirming drug stability and therapeutic efficacy over extended periods. Hamishehkar et al. investigated the potential interactions between insulin and the polymer matrix in insulin-loaded PLGA microspheres using FTIR. The spectra revealed characteristic amide I and II bonds at approximately 1655 cm^−1^ and 1540 cm^−1^, indicating the protein structure and preservations of these peaks in formulations and suggesting that no significant chemical interaction occurred with the polymer matrix. Such findings are essential as they confirm that the encapsulation process does not compromise the drug’s integrity, thereby ensuring a consistent drug release profile [229].

### 4.6. X-Ray Diffraction

This technique is used to determine the amorphous or crystalline nature of the drug. The nature of the drug in formulation is compared against that of pure drug diffractograms to investigate the polymorphic forms of matrix materials [230]. As the dissolution profile of the drug is known to be affected by the polymorphic form, determining its crystalline or amorphous nature needs to be confirmed by XRD and FT-Raman [231]. For veterinary drug delivery, confirming the physical state of the drug is important, as it influences drug solubility, dissolution rates, and release kinetics. For example, in the study by Zhao et al., minocycline was loaded into a PLGA matrix for controlled release in a veterinary model of periodontitis. XRD analysis revealed that a sharp crystalline peak of the original minocycline HCl was absent in the minocycline-loaded PLGA drug delivery system. This transformation from the crystalline to the amorphous state was critical in achieving the sustained 30-day release profile necessary for antibacterial treatment. Such findings also confirm the encapsulation and molecular dispersion of drugs in the polymeric matrix [232].

### 4.7. Encapsulation Efficiency (EE)

The encapsulation efficiency of nanoparticles and microspheres is assessed using centrifugation or ultrafiltration. To determine this, a known quantity of the formulation is dispersed in an organic solvent that dissolves the drug, and the mixture is then centrifuged and filtered. The filtrate is diluted with a mobile phase and analyzed using HPLC or a UV spectrophotometer [223,233]. Encapsulation efficiency (%) can be calculated by using the following equation.$$EE\%\;=\frac{D\;(amount\;of\;drug\;in\;the\;formulation)}{Dt\;(total\;amount\;of\;drug\;added)}\times 100$$

In LAIs for veterinary purposes, achieving a high EE is particularly important due to the need for less dosing to reduce the handling stress for animals, and for economic feasibility for large animal groups. This is one of the key characteristics to be evaluated for all the formulations, especially the ones prepared using polymers, since drugs will be encapsulated inside the polymer matrix. The extent of EE varies depending on the fabrication technique and the physio-chemical properties of the drug. Understanding EE in the drug delivery system aids in accurate dosing [228,232].

### 4.8. In Vitro Drug Release Studies

These studies are designed to evaluate how the drug is released from a dosage form into a controlled environment simulating the body conditions. It is essential to understand the release mechanism and rate of drug release. The commonly used and reported in vitro release methods are the dialysis tube method, sample-and-separate method, and continuous flow cell method, which are discussed in detail in the following sections [221,230].

### 4.9. Syringeability

For the formulation to be accepted as injectable, it needs to be easily syringeable. Syringeability testing involves the measurement of the ejection force of the formulation from the syringe via a needle to the injection site. Instruments such as a texture analyzer are used for syringeability testing [234]. Syringeability or injectability is a critical parameter in the development of LAIs for veterinary applications, especially to reduce stress while administering. The European Medicines Agency (EMA) emphasizes the importance of syringeability in veterinary medicine products, stating that it should be demonstrated concerning viscosity, particle size, and formulation homogeneity [235]. The study conducted by Xin et al. revealed that for high molecular weight compounds, high viscous solutions, and high concentration suspensions, understanding the syringeability is very crucial as it helps in determining the needle gauge, length, and size of the syringe to deliver the right amount of dose [236]. In a study by Yu et al., a PLGA-based florfenicol LAI was developed for pigs and cattle. The study reported that optimal syringeability was measured as injection force of <20 N through an 18 G needle, which was achieved when the particles were below 50 µm and in formulations containing low-viscosity vehicles such as benzyl benzoate [237].

### 4.10. Sterilization

Numerous sterilization techniques have been developed, such as moist and dry heat sterilization, filtration, and irradiation, along with some emerging techniques, such as plasma treatment and supercritical fluid treatment; however, the application of these techniques for sterilizing long-acting injectable products is limited and varies from case to case. The terminal sterilization method is preferred for sterilizing the final product, as most long-acting products cannot be filtered [238]. Dry heat and steam are the most commonly used techniques for terminal sterilization; however, this method is not suitable for PLGA microspheres or implants due to the instability of the polymer at high temperatures [239]. In such cases, gamma and X-ray irradiation are preferred [240]. Another alternative method includes e-beam (β irradiation) which is limited to the terminal sterilization of proteins and polymeric implants due to its limited penetration potential [241]. Other methods utilized for liquid formulations such as nanosuspensions include the sterile filtration method, where there is a restriction on the particle size of the suspended phase, which stipulates that 100% of the particles should be below the 200 nm range [242]. Formulations that cannot be sterilized by heat sterilization or sterile filtration need to be formulated in a closed aseptic environment [243]. The overall characterization techniques required for each drug delivery system are summarized in Figure 2.

## 5. In Vitro Drug Release Testing of Long-Acting Formulations and Its Challenges

Long-acting formulations continuously release drugs over an extended period [244,245]. This sustained release, combined with drug elimination, results in a relatively stable concentration of the drug at the target site [246]. The release profile of the drug is critical in determining the pharmacological response, with an initial burst release often employed to provide a rapid dose for the immediate management of a specific condition or disease [247]. Moreover, the burst release of a drug, mean release time, and rate of release directly affect both the duration of treatment and maximum concentration achieved in the body [246]. To ensure the effective and predictable behavior of sustained-release formulations, including long-acting formulations, a thorough understanding of the in vivo drug release kinetics is essential. However, testing each formulation in vivo during the early development stages is impractical due to time constraints, costs, and ethical considerations regarding animal use. To overcome this challenge, substantial research efforts have been directed toward developing predictive in vitro release models that can serve as reliable tools for ensuring product performance and consistency between batches [248,249].

In vitro release techniques are generally straightforward to implement and should ideally forecast the in vivo release profile. Furthermore, these methods must be capable of distinguishing between batches that meet specifications and those that do not, while also demonstrating complete or greater than 80% drug release [250,251]. At present, no standardized in vitro release testing protocol exists for long-acting formulations. As a result, researchers have employed a wide array of methods, including a United States Pharmacopeia (USP) apparatus designed for alternative administration routes and custom-developed approaches. The in vitro release techniques used for parenteral products can be broadly categorized into three groups: dialysis, sample-and-separate, and continuous flow methods. Despite their advantages, each method has its limitations. Moreover, hybrid approaches such as combining dialysis with flow-through methods have been developed [246]. To choose or design an appropriate dissolution apparatus, it is essential to thoroughly understand the drug release mechanism, as significant differences in release profiles have been observed between different apparatuses [252].

Due to the complexity of some parenteral drugs, a tailored approach will be adopted for in vitro release testing, rather than a one-size-fits-all strategy. However, presenting general approaches and options will still be beneficial. The in vitro test environment should mimic the key aspects of the intended physiological conditions, including pH, osmolarity, and buffer capacity. In some instances, non-sink conditions may also prove informative [253]. For some parenteral dosage forms, the drug release test might require the use of modified equipment, either from within or outside the compendium. For example, it may be appropriate to use varying volumes of dissolution medium, with or without agitation. Incubation techniques and the use of dialysis membranes have shown promise for some injectable microsphere formulations [253].

### 5.1. Sample-and-Separate Method

The most widely used technique for assessing the release profile of long-acting formulations is the sample-and-separate method, owing to its ease of use and practicality. This approach involves suspending micro- or nanoparticles in a specific volume of release media and agitating the system. Samples are taken at different intervals, filtered, or centrifuged to separate the particles from the media, and the media is replaced in the dissolution vessel (Figure 3). In vitro release studies involve adjusting several parameters for effective sample analysis, and one such crucial parameter is agitation. To simulate the physiological flow rates, agitation speeds are typically achieved at a speed of 100 rpm. Various methods, such as magnetic stirrers, shaking water baths, and rotating bottles, can be used for agitation [254,255,256].

The media volume selection depends on drug solubility and apparatus compatibility to maintain sink conditions throughout the study. Sample separation techniques involve filtration or centrifugation, both of which have drawbacks such as labor intensiveness, potential interactions with filter materials, and sample loss. Alternatively, some studies allow particulates to settle or perform in situ analysis without separation. Sink conditions and the sensitivity of quantification techniques influence the sampling volume. However, standardized methods requiring large media volumes might not be practical for small-volume injectable forms [257,258]. The sample-and-separate technique offers a straightforward and relatively precise evaluation of drug release patterns in vitro. Nevertheless, this approach has certain drawbacks. For instance, insufficient stirring can lead to the clumping of microspheres during extended in vitro release tests, as well as the potential loss of the drug formulation when sampling is performed.

### 5.2. Continuous Flow

In vitro release testing has employed continuous flow release techniques to mimic in vivo conditions [259]. The continuous flow apparatus comprises several components: a flow-through cell containing the sample, a filter positioned at the top of the cell to prevent particle escape, a pump driving dissolution media through the cell, a water bath maintaining media temperature, and a media reservoir. This setup can be arranged in either closed- or open-loop configurations (Figure 4). The closed-loop system recirculates the same media continuously, whereas the open-end configuration incorporates a sample collector, allowing fresh media to pass through the cell only once [260].

The continuous flow method involves manipulating specific parameters to optimize the in vitro release testing of pharmaceutical formulations. One important aspect is sample immobilization, where the use of glass beads helps to position various formulations such as tablets, implants, and microparticles. These glass beads serve two purposes: maintaining laminar flow conditions and preventing issues such as microsphere aggregation or incomplete release profiles. Currently being investigated for standardizing in vitro release testing of microspheres, the incorporation of glass beads aims to enhance testing accuracy and consistency [261]. Another crucial factor to consider is the flow rate, which determines the continuous flow within the system through pumps, such as peristaltic, syringe or high-performance liquid chromatography (HPLC) pumps [259,261,262]. Flow rates ranging from 0.4 μL/mL (HPLC pump) to 200 L/h (peristaltic pump) can be covered with the pumps. The extent and kinetics of drug release, particularly for diffusion-controlled release systems, are affected by the flow rate. By influencing material hydration and drug diffusion, the flow rate can impact the speed and completeness of the release process. Lower flow rates may result in slower and incomplete release due to inadequate media transfer, underscoring the importance of selecting an appropriate flow rate for accurate testing outcomes. In the case of an open-loop configuration, the flow rate represents drug clearance from the release site, which is at subcutaneous or intramuscular sites for long-term implants and injections [263].

Media recycling is crucial in the continuous flow method, particularly when using a peristaltic pump. Using a closed-loop setup, the recycled media is redirected back to the sample chamber. This approach is more practical in terms of both sampling convenience and the overall media volume required if the conditions for maintaining sink conditions are met. When considering media recycling, the drug’s solubility must be considered, similar to other release testing methods. Alternatively, the option of using the fresh buffer and pumping it through the flow-through cell (open loop) is also available, offering flexibility in the experimental setup [246]. While the CF method has streamlined routine sampling and media replacement procedures, it is not without its drawbacks. These limitations include high equipment expenses, complex setup processes, filter blockage issues, adherence to filters and glass beads, and challenges in maintaining steady flow rates. Consequently, these factors contribute to the significant variability in experimental outcomes [264].

### 5.3. Dialysis Method

This technique involves isolating a sample from the main medium by placing it in a dialysis sac (Figure 3). The sac is a porous cellulose membrane with a varied range of molecular weights, allowing the released drug to pass through and enter the main medium for analysis [248,265]. It is standard practice to utilize membranes with molecular weights that are ten times higher than the released compound. This ensures that the drug can freely diffuse through the membrane, resulting in equal drug concentrations inside and outside the dialysis sac. The internal volume of the dialysis bag is considerably smaller than the surrounding medium. To aid drug diffusion, the volume of the inner medium is kept at five to ten times less than that of the outer medium. This volume difference generates the necessary force for drug transfer to the external environment and maintains sink conditions. Additionally, the bulk media should be stirred using techniques such as a magnetic stirrer, paddle, or horizontal shaker. In general, dialysis can be classified into two categories: regular and reverse dialysis [266].

In regular dialysis, the samples are inserted into a dialysis sac, allowing drug diffusion across the entire sac wall, or are placed in a tube with a dialysis membrane at one end, resulting in a reduced drug diffusion surface area (Figure 3). Despite debates surrounding the suitability of dialysis for in vivo-relevant release testing, it is hypothesized that this method may accurately predict in vivo behavior for samples that are essentially immobile and can be enveloped by a static membrane, such as subcutaneous and intramuscular injections [267]. Reverse dialysis works on a similar principle but with the sample located in the bulk media rather than the dialysis sac. Media-filled dialysis sacs are immersed in a bulk container, and sampling is performed by either opening the sac and extracting a portion of media or removing the entire sac and substituting it with a new one. This approach offers the benefit of agitating the microparticle suspension instead of the bulk media, thereby preventing aggregation and promoting polymer hydration and drug diffusion. This results in more consistent outcomes and addresses the issue of violating the sink conditions previously noted with the dialysis technique. However, the gradual equilibration process with the external medium impedes accurate initial drug concentration measurements [268]. Furthermore, the DM method has several drawbacks, including the challenge of maintaining sufficient agitation to prevent microparticle clumping inside the dialysis bag, the unsuitability of drugs that adhere to the polymer or dialysis membrane, and the potential violation of sink conditions [266,269].

### 5.4. Standardized Methods

The USP prescribes standardized methods for testing the dissolution of various pharmaceutical formulations, including both immediate and controlled-release formulations. These methods make use of different apparatuses, each intended for specific purposes. Some of these apparatuses include USP Apparatus 1 (basket), Apparatus 2 (paddle), Apparatus 3 (reciprocating cylinder), Apparatus 4 (flow-through cell), Apparatus 5 (paddle over disk), Apparatus 6 (cylinder), and Apparatus 7 (reciprocating holder). Apparatus 1 consists of a basket immersed in dissolution media that is rotated to test the solid oral dosage forms [270]. Apparatus 2 employs a motor-driven paddle to disperse samples, making it a commonly used reference point for new dissolution tests and release kinetics comparisons. Apparatus 3, featuring an inner vessel for the sample and an outer vessel for sampling, is not widely favored for in vitro release testing due to concerns about media evaporation. Apparatus 4, known as the flow-through cell, is widely used for controlled release parenteral formulations, utilizing a pumping mechanism to pass dissolution media through a cell that holds the sample. Apparatuses 5 and 6 are designed for transdermal delivery systems, using a disk assembly or a metallic cylinder to immobilize and test these systems. Apparatus 7, a modification of Apparatus 3, is adapted for testing transdermal, osmotic devices and stents by incorporating a holder to carry the implant [260]. There has been a significant focus on Apparatus 4 due to its versatility and success in the in vitro release testing of controlled-release parenteral formulations. Researchers are working on developing standardized protocols to ensure consistent and repeatable results across a variety of formulations.

### 5.5. Accelerated In Vitro Dissolution Testing Methods for Long-Acting Parenteral Products

Accelerated in vitro release methods have drawn significant attention for studying drug release, as they shorten the time needed to perform the tests [271]. Several parameters, including pH, temperature, solvent, surfactants, and agitation rate, have been employed to achieve accelerated release. Nevertheless, it is important to note that the accelerated conditions may not only change the rate of drug release but may also affect the mechanism of drug release, underscoring the need to not only understand the drug release mechanism but also how the accelerated parameters affect the release mechanism [272,273,274]. Ideally, drug release from accelerated and “real-time” tests should follow the same release mechanism, but a change in the drug release mechanism is acceptable as long as the accelerated release method enables the effective discrimination of batches and demonstrates a similar rank order relationship of different formulations compared to the real-time release method [240]. It is advised that the specifications for accelerated release should include determining at least 80% of the cumulative amount to be released for comparison with ‘real-time’ studies [272,275].

Elevated temperature is a widely used approach to accelerate the release of drugs from long-acting formulations, particularly from polymeric-based formulations [275,276]. High temperature increases polymer mobility, hydration, and degradation rate, resulting in increased drug release. Generally, it is advised to control the elevated temperature below the glass transition temperature (Tg) of the polymers, as the release mechanism might be altered at temperatures above Tg. Accelerated tests using elevated temperatures are effective in predicting real-time release for erosion-controlled systems [269]. Also, the need to complement the accelerated test with the real-time release for the initial burst release is recommended due to the observed change in morphology of the formulations because of plasticization under elevated temperatures, which decreases the release of drugs [277].

The pH level of the surrounding environment also affects the hydrolytic degradation rate of biodegradable polyesters like PLGA, thereby influencing drug release patterns. Both acidic and basic conditions can accelerate the degradation of PLGA [278]. In acidic environments, PLGA primarily experiences bulk erosion, showing degradation behavior similar to that at physiological pH (7.4), but with more uniform morphological changes. Conversely, in highly basic conditions (pH > 13), PLGA degradation transitions to a surface erosion mechanism [279,280]. While extreme pH levels can speed up drug release, their overall effect is generally less significant than that of increased temperatures, and they may not be appropriate for drugs that are unstable in such conditions [274]. The addition of surfactants or organic solvents into the release media has also been employed to accelerate drug release from long-acting parenteral formulations. The addition of surfactants such as Tween 20 to the release media of lipid implants can facilitate wetting and increase buffer penetration. It also increases drug solubility in the release media, thereby increasing the rate of drug release. Some surfactants, on the other hand, form cracks in the lipid matrix by interacting with the lipid matrix, resulting in increased drug release [281,282].

Similarly, the addition of organic solvents such as ethanol and acetonitrile to the release media has been successfully used to achieve accelerated drug release by increasing the porosity of PLGA [281]. Several attempts have been made to integrate accelerated conditions with the commonly used in vitro release testing methods for long-acting injections. For example, accelerated methods based on the sample-and-separate methodology at an elevated temperature (50 °C) and acidic pH (4) were evaluated to increase the release of a drug (leuprorelin) from a depot formulation based on PLGA depot formulation [283]. The accelerated release correlated well with the real-time release of the drug and was able to discriminate between different formulations. Similarly, elevated temperature and acidic pH-accelerated test conditions using USP 4 were developed. Shen and colleagues developed a reproducible accelerated in vitro release method for long-acting PLGA microspheres with inner structure/porosity differences using risperidone as a model drug. Among the evaluated methods, the accelerated USP 4 method was found to be reproducible and able to discriminate the release profile among formulations with different porosities [284].

## 6. In Vitro and In Vivo Correlation

The FDA’s IVIVC guidance document describes IVIVC as a predictive model that illustrates the relationship between an in vitro characteristic, typically the extent or rate of drug dissolution or release, and a pertinent in vivo response, such as the concentration of the drug in plasma or the quantity absorbed. IVIVC’s primary function is to accurately forecast a product’s expected bioavailability based on its dissolution profile. It can be utilized to request a biowaiver, eliminating the need for bioequivalence studies in favor of in vitro release testing. However, the FDA generally rejects biowaiver applications when substantial changes have been made to the formulation, including alterations in dosage strength. To obtain a biowaiver in such cases, multiple IVIVCs for various dosage strengths must be established and meet specific criteria [251].

For more than 20 years, researchers have relied on this guidance for developing IVIVCs for formulations beyond oral dosage forms. An industry survey on IVIVC development revealed that half of the respondents seldom or never created IVIVC/IVIVR models for non-oral dosage forms, underscoring the difficulties associated with these formulations [285]. Among the FDA-approved LAI aqueous suspension products, Invega Sustenna is the only product that has demonstrated a clinical level A IVIVC based on the formulations with varying particle sizes. Recently, animal models have been utilized for IVIVC model development in non-oral dosage forms to mitigate the high costs and challenges of clinical studies. These animal studies may facilitate the successful development of human IVIVCs. For instance, Level A (point-to-point) IVIVCs have been successfully developed for various microspheres [286,287,288,289]. While the FDA has not released specific IVIVC guidelines for parenteral products, the principles from their guidance for extended-release oral dosage forms have been adapted for parenteral drugs. IVIVC encompasses five categories: level A, B, C, multiple C, and D. The FDA guidance document, however, only elaborates on levels A, B, C, and multiple C, as level D is simply a rank-order comparison [193].

Level A IVIVCs are the most comprehensive and informative, making them the FDA’s preferred type of correlation. They establish a point-to-point relationship between in vitro and in vivo release profiles that can be either linear or nonlinear, requiring appropriate modeling for nonlinear cases [290]. Level B IVIVCs, in contrast, are the least predictive, often relying on statistical analysis to compare the mean in vitro dissolution time with the mean in vivo dissolution time or mean residence time. However, their limited predictability stems from the fact that different in vivo release profiles can produce the same mean dissolution or residence time. Despite this limitation, level B correlations can estimate the overall in vivo release duration, which is particularly relevant for long-acting implants and injections [291]. Level C IVIVCs establish a single-point correlation between an in vitro release parameter (such as disintegration time, dissolution rate, or the time required for a specific percentage of drug release) and pharmacokinetic parameters (such as Cmax, Tmax, dissociation constant, time for a certain percentage of drug release, or AUC). While less informative than level A IVIVCs, multiple level C IVIVCs, where multiple in vitro dissolution parameters correlate with multiple pharmacokinetic parameters, may be useful. However, if multiple level C IVIVCs can be established, achieving a level A IVIVC is often possible and preferred [292]. Level D IVIVCs, also known as rank-order correlations, compare the relative release rates of a drug in vivo and in vitro. Since they provide only qualitative insights, they are not considered useful for regulatory purposes [293].

Only a few studies have reported establishing an IVIVC of long-acting formulations in animals. For example, Larsen et al. determined the in vitro release of subcutaneously administered bupivacaine oily solution using the rotating dialysis cell model: A complete release of the drug was observed in 50 h. First-order kinetics described the release profile of the drug very well. Similarly, the in vivo release kinetics was first-order and corresponded well with the in vitro release kinetics found using a rotating dialysis cell [294]. The IVIVC of leuprolide released from an implantable device was also reported. A level A IVIVC with excellent correlations (R^2^ of 0.99) was established. The in vitro release experiment was conducted using an in-house flow cell apparatus containing a chip assembly mounted in a flow cell. The in vivo study was conducted by implanting the device in the subcutaneous tissue of male beagle dogs. The PK data were also used to calculate fractional absorption by the application of the Wagner–Nelson equation [295].

Schliecker and colleagues also reported the development of in vitro drug release and in vivo pharmacokinetic parameter correlations for an implant containing Buserelin in beagle dogs. Theoretical models of Korsmeyer–Peppas and Higuchi were used to analyze the in vitro release profile of the drug. Level B correlation was established between the in vitro dissolution time and in vivo residence time without considering the drug release mechanism. However, for formulations with a predominant diffusion level, a level A IVIVC was established [296]. As can be observed, only a few studies reported the direct correlation between in vitro release and in vivo release profiles, demonstrating the difficulties in establishing the IVIVC for these formulations. Hence, further studies that explore in vitro models that mimic in vivo conditions are required.

## 7. Physical and Chemical Stability of Long-Acting Drug Delivery Systems

LAIs have emerged as a transformative approach in drug delivery, offering the sustained release of active pharmaceutical ingredients over days and weeks to months. Ensuring their physical and chemical stability is crucial as it directly impacts the safety and efficacy of the formulation [240]. Physical stability ensures that the formulation maintains its physical characteristics such as particle size distribution (PSD), morphology, and crystallinity over time. Physical instability, such as particle aggregation, sedimentation, phase separation, or crystallization, can lead to dose variability, inconsistent absorption, or needle clogging. PSD is a critical factor for drug delivery forms which include nanosuspensions, nanocrystals, liposomes, and microspheres. To enhance the physical stability, stabilizers, emulsifiers, or suspended solids can be added if required to maintain the integrity of the particles. Nandi et al. developed a nanosuspension of itraconazole for sustained release, and with the inclusion of vitamin E TPGS, the particles were stabilized and were thermodynamically stable for up to 500 days [297]. Also, certain formulations have a high concentration of the suspension of a drug or polymer that can pose a risk of needle clogging due to particle bridging; for such formulations, ensuring particles are freely dispersible and not coagulated is necessary to prevent clogging of needles while administering [298].

Chemical stability refers to the resistance of the drug and formulation components to degradation under various conditions such as temperature, pH, and light. Achieving chemical stability is critical to ensure the safety and potency of LAIs over their shelf-life and in vivo release performance. Formulations that include pH-sensitive or photo-sensitive drugs or proteins are highly prone to degradation via hydrolysis or oxidation. To mitigate these factors and enhance the chemical stability of the formulation, the quality by design (QBD) approach can be employed by identifying the critical material attributes (CMA) and critical process parameters (CPP) [299]. The use of biodegradable polymers, antioxidants, or suitable stabilizers can minimize the risk of chemical degradation and increase long-term stability [300]. To understand the stability profile of the formulations, stability studies as per the ICH guidelines can be performed, and accelerated stability studies, if performed in the early development stages, can provide a better idea about and aid in developing a stable drug delivery system [301,302].

## 8. Safety, Biodegradability, and Biocompatibility Considerations of Long-Acting Drug Delivery Systems

It is well-established that long-acting formulations have a significant advantage owing to their ability to provide a steady and consistent dose of medication without the need to administer multiple doses. However, there are still some concerns related to the safety, biodegradability, and biocompatibility of these long-acting formulations, potentially deterring their application in veterinary health. Very little research has been conducted to address the impact of these factors on the acceptability of LAIs in animals. However, studies conducted on humans can be used to shed some light on these potential issues. In humans, the prescribing rate of LAIs is low, mostly due to the fear that they might cause serious adverse events and greater risks of adverse effects [303]. This can be explained because LAIs are initially administrated in large single doses, and in case of the occurrence of any serious adverse events, the therapeutic agents cannot be discontinued immediately [303].

In a study conducted by Lehman et al., the clinical safety and pharmacokinetics of a novel long-acting injectable omeprazole (LAO-USA) in treating equine gastric ulcer syndrome (EGUS) were evaluated. The study also determined that the injection site reactions of the first two doses were 8% and 13% of the starting dose of 5 mg/kg, gradually increasing to 22% and 48%. Despite this observation, the exaggerated reaction after the increased doses might have been due to the sensitization of the neck area (the site of injection) to the drug after receiving the first two doses, rather than because of the formulation itself. Although there were statistically significant changes in complete blood count (CBC) and serum biochemistry (SBP), these values remained within the normal reference range. A decrease in body weight was also observed; however, it was likely caused by the increased workload of these horses due to their on-site training during the study period. The primary drawback of this safety study was the lack of control subjects, making the data inconclusive [304]. Similarly, Gruen et al. conducted a randomized, placebo-controlled, double-blind study to evaluate the safety and efficacy of frunevetmab at monthly intervals to treat osteoarthritis (OA) pain in cats. Most adverse effects were unrelated to treatment and were observed in both treatment and placebo groups. However, collective skin-related adverse events occurred significantly more frequently in frunevetmab-treated cats. In most cats, the skin-related adverse events were related to traumatic injuries or a history of allergic dermatitis [305]. However, further work is needed to better understand the reasons for this.

In contrast, several other studies have revealed the opposite of the safety concerns of LAIs, where the lower peak-to-through blood level variations in LAIs are associated with fewer adverse effects [306]. For example, in the recently published study containing a 16-year case series involving Hong Kong citizens diagnosed with schizophrenia, long-acting injectable antipsychotics were associated with a lower risk of adverse events, fewer hospitalizations, and fewer suicide attempts compared to oral antipsychotics [307]. Similarly, in a comparative study conducted by Park et al. on LAIs and oral second-generation antipsychotics (SGAs) in treating schizophrenia, LAI SGAs were associated with a significantly lower relapse rate, shorter hospitalization time, and longer time to relapse [308].

In addition to safety, biodegradation and biocompatibility are crucial for developing successful long-acting formulations [309]. However, there are very few biodegradable deliveries on the market, both in terms of human and veterinary formulations. Many drugs are sensitive to temperature, shear forces, or solvents. To avoid stability problems, the use of biodegradable materials that allow the incorporation of sensitive drugs without the use of harsh treatments is required [310]. As a result, the behavior and rate at which the system degrades, and its biocompatibility during the prolonged biological residence, must be thoroughly studied. In the case of oil-based LA products, the viscosity of the oil has a significant impact not only on the release of incorporated drugs but also on the tolerability of the injections [112]. For instance, highly viscous sesame oil was shown to improve the tolerability of IM injections [112]. Benzyl alcohol, which is used as a preservative, can also be introduced to adjust the viscosity of oily vehicles. Interestingly, the in vivo clearance data of these oily formulations can differ between animals and humans due to the differences in the immune systems between species. For example, the sesame oil and benzyl alcohol with (10% *w*/*v*) combination disappeared within one week in humans following the IM injections, whereas it took 31 days in rats [311].

In the case of polymeric LAIs, the most widely employed biodegradable polymers are PLGA and PLA [312]. The degradation of these polymers refers to the cleavage of the polymer, leading to a net loss of molecular weight and the erosion of the materials [312,313]. The degree and rate of polymer degradation depend on various elements, including the copolymer ratio, crystallinity, molecular weight, and polydispersity [310]. Due to these advancements, these biodegradable polymers were employed in numerous veterinary formulations. For example, ivermectin was incorporated into bioabsorbable, injectable PLGA copolymer microspheres to control ectoparasites on livestock pests [314]. Similarly, moxidectin-incorporated PLGA microspheres were successful in the treatment and prevention of canine heartworm in dogs [315]. The formulations were safe and well tolerated in dogs, with no adverse events observed. This product is now approved and available on the market as ProHeart™ (USA and Australia) and Guardian SR Injectable™ (Italy) [315].

Various excipients, including polymers and solvents, are utilized in the development of LAI formulations for veterinary use. Commonly used organic solvents include acetone, ethyl acetate, dichloromethane, and others like dioxane, hexane, and tetrahydrofuran. While solvents such as acetone, ethyl acetate, and dichloromethane are already employed in pharmaceutical manufacturing, biocompatibility and low toxicity are essential criteria for their use in parenteral formulations [316,317]. To ensure injectability, excipients must also possess suitable solubility and allow for low-viscosity solutions. Although isotonic aqueous solutions remain the standard for injectable products, several organic solvents have been evaluated and accepted by regulatory authorities for specific applications [318,319,320]. However, the available data on their toxicity, tolerability, and systemic safety are limited or conflicting. Among the solvents explored for in situ-forming implants, N-methyl-2-pyrrolidone (NMP), triacetin, benzyl benzoate, glycofurol, and glycerol formal have received significant attention [316]. Of these, glycerol formal is the only solvent that has been approved for veterinary use, featured in Ivomec-S 0.27%™ for subcutaneous injection in piglets [321]. Studies in dogs have shown that both glycerol formal and triacetin demonstrate acceptable safety profiles [322,323]. Conversely, NMP has raised concerns due to its classification by the European Medicines Agency (EMA) as a Class 2 solvent, indicating potential risks such as neurotoxicity and teratogenicity [324]. Although mild local tissue reactions were reported in Rhesus monkey studies [58], NMP use in dogs and cats has been associated with pain upon injection and inflammatory responses, rendering it unsuitable for veterinary applications [316].

While adjuvants are essential in enhancing immune responses, especially in vaccine formulations, their interaction with antigens can sometimes trigger undesirable effects. These may include both systemic and local immune-mediated reactions. The systemic and non-specific adverse effects reported following adjuvant use included fever, lethargy, anorexia, arthritis, uveitis, and soreness [325,326,327]. In some cases, adjuvants have been implicated in autoimmune phenomena. For example, overdoses of interleukin-2 (IL-2), proposed as an immunostimulatory adjuvant, have been associated with the development of autoimmune diseases [328]. Additionally, a temporal association has been observed between the administration of certain canine vaccines—such as those for distemper, rabies, and parvovirus—and the appearance of autoantibodies or autoimmune hemolytic anemia [326]. Adjuvants may also elicit adverse effects specific to their chemical composition. Crude saponin-based adjuvants, for example, are known to cause hemolysis if administered intravenously [329]. More commonly, adjuvants are associated with local injection site reactions, including inflammation, granuloma formation, and in rare cases, sterile abscesses. In companion animals, rabies and distemper combination vaccines are most frequently linked to these local effects in dogs, while rabies vaccines are more commonly associated with such reactions in cats [330]. Though generally minor and self-limiting, these local reactions can sometimes have significant consequences.

In studies evaluating oxytetracycline (OTC) formulations in calves, pigs, and sheep, both conventional and long-acting injectable products demonstrated notable local adverse effects. Specifically, three long-acting formulations with 20% drug loading and one conventional formulation with 10% loading induced significant muscle irritation at the injection site when assessed 10 days post-administration. Moreover, oxytetracycline residues were detected in the organs and at the injection sites in all but one conventional formulation tested. These variations in tissue irritation and drug residue profiles were strongly associated with the different solvent systems used in the formulations [331]. In a related study, the extent of tissue irritation directly impacted the recovery of OTC from the injection site, with the most pronounced effects observed in the long-acting 20% formulation. This suggests that excessive local inflammation not only affects tissue health but can also impair drug release and bioavailability, potentially compromising therapeutic outcomes [21].

Systemic adverse effects have also been reported with other long-acting drugs. High concentrations of doxycycline—equivalent to 8–16 times the average MIC—may exhibit concentration-dependent antibacterial effects in vitro [332,333]. However, overdoses 5–10 times higher than the recommended dose have resulted in cardiac toxicity in calves, with plasma concentrations ranging from 50 to 100 μg/mL. These findings highlight the risks associated with achieving excessively high Cmax values and reinforce the importance of cautious dose optimization for sustained-release formulations [334,335]. Additionally, mild injection site reactions have been observed in dogs administered with a five-fold overdose of microsphere-based formulations, including minor swelling and temporary suppression of erythropoiesis. Although hematological parameters remained within normal limits and no clinical symptoms were observed, these findings underscore the need for dose control and careful monitoring [336]. A comparison of adverse event rates from clinical studies provides further insight into the safety profile of LAFs. In an observational study by McTier et al. [337], a high proportion of dogs (87.9% and 85.1%) experienced at least one side effect after the administration of ProHeart^®^ 12 or Heartgard^®^ Plus, respectively. Common reactions included vomiting, lethargy, diarrhea, and anorexia, with mild injection site reactions resolving within seven days. Notably, 2% of dogs experienced hypersensitivity reactions. In contrast, a separate study of Afilaria^®^ SR reported a significantly lower incidence of adverse effects, with only one case each of anaphylactoid reaction and angioneurotic symptoms among 583 dogs (0.34%) [338]. These data suggest that adverse reactions can vary widely among different LAFs, and careful formulation design can play a key role in minimizing side effects.

## 9. Conclusions and Future Prospects

From the above, it can be concluded that the advancements in LAIs in the field of veterinary medicine present numerous benefits, including reduced dosage frequency, enhanced patient adherence, and improved therapeutic outcomes, which are critical in chronic conditions. However, several challenges remain concerning safety, biocompatibility, biodegradability, and their potential to trigger adverse immune responses. Addressing these concerns will be a crucial step for the continued progress of LAIs and their acceptance among veterinary professionals and regulatory agencies. Also, a crucial factor is the necessity for a thorough IVIVC study to guarantee that these formulations deliver consistent efficacy across different animal species. Advancements in LAIs, combined with strategic collaborations between pharmaceutical companies and veterinary health organizations, have the potential to broaden therapeutic choices and enhance health outcomes in veterinary medicine.

## Figures and Tables

**Figure 1 pharmaceutics-17-00626-f001:**
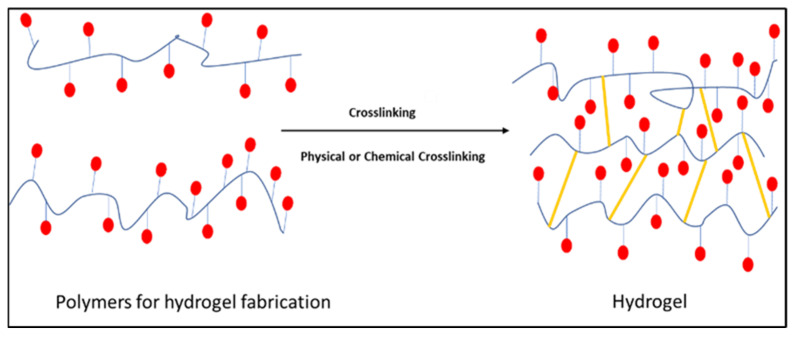
Fabrication of hydrogels.

**Figure 2 pharmaceutics-17-00626-f002:**
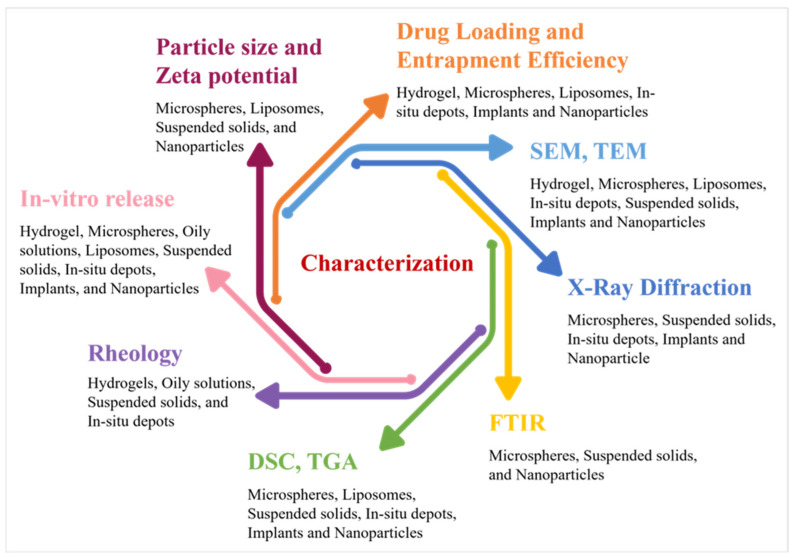
Characterization methodologies for various formulations (the template used for the image is generated using https://www.presentationgo.com/, accessed on 30 March 2025).

**Figure 3 pharmaceutics-17-00626-f003:**
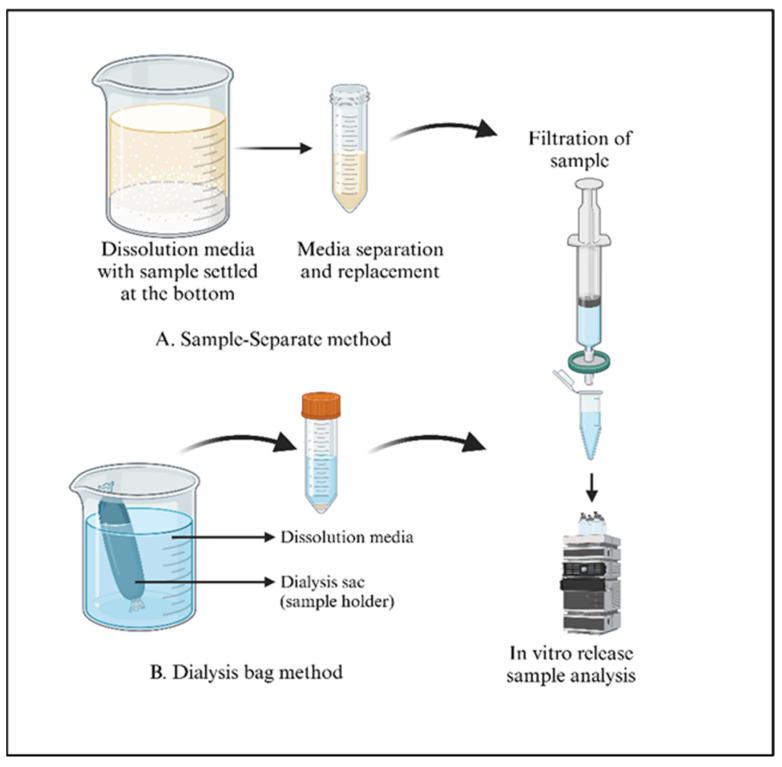
Various methods used in in vitro release studies.

**Figure 4 pharmaceutics-17-00626-f004:**
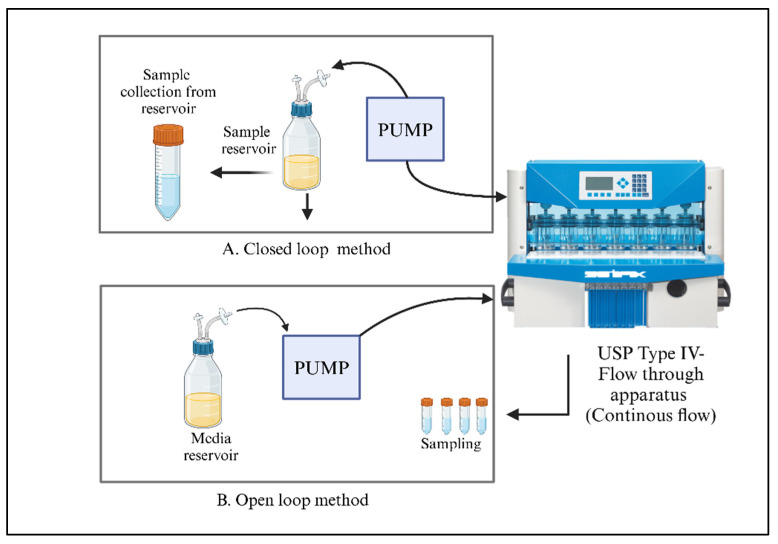
In vitro release using USP type IV apparatus: continuous flow.

**Table 1 pharmaceutics-17-00626-t001:** Different types of in situ depot-forming systems.

In Situ Depot Delivery Systems	Advantages	Disadvantages	Release Time Range	Ref.
Oily solutions	Long-term stability Ease of preparation Good injectability and syringeability properties	Only lipophilic drugs can be delivered Hydrophilic drugs need an additional carrier before they can mix with oils Spreading and pain at the site of injection	Weeks to several months	[56]
Vesicular phospholipid gels	Both hydrophilic and hydrophobic drugs and proteins can be encapsulated Organic solvent-free preparation Long-term storage	Sterilization is challenging High-viscosity solution, which requires pressure during injection	Days to weeks	[57]
Phospholipid-based phase separation gel (PPSG)	Both hydrophilic and hydrophobic drugs can be encapsulated No initial burst releases Low viscosity Can be translated for large-scale production	Ethanol must be used to solubilize drugs Ethanol residues can cause side effects	Days to weeks	[58,59]
Nanoemulsion	Both hydrophilic and hydrophobic drugs can be encapsulated	Prolonged release only lasts for hours Requires the addition of polymeric surfactants to improve stability	Hours	[60]
Liquid crystalline systems (like cubosomes and hexosomes)	Solutions transform into liquid crystals upon contact with body fluids Both hydrophilic and hydrophobic drugs and macromolecules can be encapsulated	High water content can result in the burst release of hydrophilic drugs Small changes in manufacturing can result in non-uniform crystal structures High viscosity requires high forces for injection	Days to weeks	[61,62]
Organogels	Only hydrophobic drugs can be loaded	High viscosity Organogelators can cause toxicity	Days	[63]
Hydrogels	Both hydrophilic and hydrophobic drugs and macromolecules can be encapsulated Prolonged release due to matrix system	Tedious process Requires polymers that could lower biocompatibility	Days to weeks	[64]
Polymeric microparticles	Utilization of synthetic biodegradable polymers Both hydrophilic and hydrophobic drugs and macromolecules can be encapsulated Can deliver large doses	Complex manufacturing process Burst release can be observed	Days to weeks	[65,66]

**Table 2 pharmaceutics-17-00626-t002:** List of commercially available long-acting injectables for veterinary applications.

Product Name	Active Ingredient	Key Excipients	Indication	Species	Dosing Information	Route of Administration	Manufacturer	Ref.
Oily solution								
Boostin^®^-S	Recombinant bovine somatotropin	Vitamin E acetate; lecithin	Increased milk production	Cattle	500 mg every 14 days	SC	Intervet South Africa (Pty) Ltd., Kempton Park, South Africa.	[119]
Decort 20 oily injection	Deoxycortone acetate	NA	Sodium and water retention	Dogs; cats; horses	1 mL (dogs and cats); 3–5 mL (horses) every 3 to 4 weeks	IM	Jurox Pty Ltd., Rutherford, NSW, Australia.	[120]
Depodine™	Iodine	Peanut oil	Treatment of iodine deficiency	Cattle; sheep	1.5 mL (sheep); 3–6 mL (cattle)	IM	Alleva Animal Health, Australia	[121]
Oily Suspension								
Excenel^®^ RTU	Ceftiofur hydrochloride	Phospholipids; sorbitan oleate; cottonseed oil	Bovine respiratory disease; acute bovine interdigital necrobacillosis; acute metritis	Swine	3–5 mg/kg body weight	SC; IM	Zoetis, Parsipanny, NJ, USA	[122]
Bimoxyl™ LA	Amoxicillin trihydrate	Glycerol monocaprylate; propylene glycol dicaprylocaprate	Amoxicillin is susceptible to bacterial infections; respiratory infections; urinary tract infections	Cattle; sheep; pigs; dogs	15 mg/kg body weight; repeat at 48 h intervals if required	IM (cattle, sheep, and pigs); SC (dogs)	Bimeda Animal Health Ltd., Dublin, Ireland	[123]
Moxylan LA	Amoxicillin trihydrate	Plant oil	Amoxicillin is susceptible to bacterial infections	Cattle; sheep; pigs; dogs; cats	15 mg/kg body weight; repeat at 48 h intervals if required	IM (cattle, sheep, and pigs); SC (dogs and cats)	Jurox Pty Ltd., Rutherford, NSW, Australia	[124]
SMARTSHOT^®^ B12	Vitamin B12	Peanut oil; poly(lactide-*co*-glycolide)	For the treatment and prevention of cobalt deficiency	Cattle; sheep	0.5 mL (lambs for docking); 1 mL (lambs for weaning); 5 mL (ewe); 1 mL per 25 kg live weight (calves)	SC; IM	Virbac New Zealand Ltd., Hamilton, New Zealand	[125]
POSILAC™	Recombinant bovine somatotropin	Sesame oil	To increase the production of milk in lactating cows	Cows	500 mg every 14 days	SC	Union Agener, Inc., Augusta, GA, USA	[126]
Microspheres								
Micotil^®^	Tilmicosin	NA	Bovine respiratory disease	Cattle, lamb	10 mg/kg	SC	Elanco, Indianapolis, IN, USA	[127]
ProHeart^®^12	Moxidectin	Hydroxypropyl methylcellulose	Heartworm disease	Dogs	10 mg	SC	Zoetis, Parsipanny, NJ, USA	[128]
Implants								
Component E-C and Component E-C with Tylan	Progesterone 100 mg, estradiol benzoate 10 mg, tylosin tartarate 29 mg	NA	Improves body mass gain	Beef calves	Single dose for 100–140 days	SC	Elanco, Indianapolis, IN, USA	[129]
Compudose 100, 200, and 400	Estradiol	NA	Improves body mass gain	Beef calves	Single dose for 170–400 days	SC	Elanco, Indianapolis, IN, USA	[130]
Ralgro	Zeranol 36 mg	NA	Improves body mass gain	Cattle	Single dose for 70–100 days	SC	Merck & Co, Rahway, NJ, USA	[131]
Synovex ONE Grover	Trenbolone acetate 40 mg, estradiol benzoate 21 mg	Lipid matrix	Growth	Cattle	Single dose for 200 days	SC	Zoetis, Parsipanny, NJ, USA	[132]
Suprelorin	Deslorelin acetate 4.7 mg	Lipid matrix	Contraception	Male dogs	Single dose for 6 months	SC	Virbac New Zealand Ltd., Hamilton, New Zealand	[133]
Hydrogels								
Synovetin OA Device	Tin-117 (radioisotope)	NA	For synovitis and chronic canine elbow pain	Dogs	Once a year	IA	Exubrion, Gainesville, GA, USA	[134]
Ivermectin hydrogel	Ivermectin 5 mg/mL	Propylene glycol mono myristyl ether propionate	Antibacterial and antiparasitic infections	Cattle	14–21 days	NA	International Animal Health Products, Huntingwood, Australia	[135]
Conveina^®^	Cephalosporin 80 mg/mL	NA	For periodontal diseases	Dogs and cats	14 days	SC	Zoetis, Parsipanny, NJ, USA	[136]
Doxirobe^®^ Gel	Doxycycline hyclate	PLGA	For periodontal diseases	Dogs	7 days	IPD	Zoetis, Parsipanny, NJ, USA	[137]

IM: intramuscular, SC: subcutaneous, IA: intra-articular, IPD: intra-periodontal; NA: Not available.

**Table 3 pharmaceutics-17-00626-t003:** Preclinical, pilot, and clinical studies reported for liposomal drug delivery in animals.

Drugs	Animals	Study Details	Ref.
Stealth PEGylated liposomes			
Doxorubicin	Dogs	Randomized efficacy and toxicity studies The dose was administered via IV every 3 weeks Partial responses were observed with cutaneous toxicity and no cardiomyopathy or neutropenia	[181,182]
Doxorubicin	Dogs	Prospective, uncontrolled, and unmasked toxicity and pharmacokinetic efficacy studies Intraperitoneal administration compared to IV administration	[183]
Doxorubicin/Caelyx	Cats	Efficacy studies 70% response rates were observed	[184]
Non-PEGylated liposomes			
Doxorubicin/Myocet	Dogs	Preclinical toxicity, efficacy, and case reports IV every 3–6 weeks for chemotherapy-resistant myeloma	[185,186]
Liposomes			
Doxorubicin	Dogs	Toxicity and pharmacokinetic studies Severe adverse effects were observed	[174]
Carmustine	Dogs	Biodistribution, safety, and proof of concept studies Study ongoing	[187]
Vincristine and paclitaxel	Dogs	Pharmacokinetic and biodistribution studies An increase in the therapeutic index was observed in liposomal vincristine after a single injection of IV A 15-fold higher concentration of paclitaxel was observed with the liposomal formulation than with a free paclitaxel IV	[188,189]
HAS cell lysates	Dogs	In a pilot study for canine hemangiosarcoma	[190]
Inactivated avian pathogenic *E. coli*	Chickens	In a pilot study for the avian colibacillosis vaccine	[191]
MIC3 protein from *T. gondii*	Sheep	In a pilot study for the *Toxoplasma gondii* vaccine	[177]

## Data Availability

No new data were created or analyzed in this article.

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
