# Peer review of "Unveiling the Future: Opportunities in Long-Acting Injectable Drug Development for Veterinary Care"

_pharmaceutics, 2025, doi:10.3390/pharmaceutics17050626_

Round 1

Reviewer 1 Report

Comments and Suggestions for Authors

Overall, this review is very comprehensive and thorough, covers a lot of the LAI products for veterinary care.  In Section 4.10 Sterilization, the authors omitted the use of e-beam to sterilize the product.  In section 5, on in vitro release, the authors describe the techniques as generally straightforward to implement, which may be true but the issue with in vitro release is that the assays take a long time to run, since you generally want real time release data (and these formulation are, after all, long-acting.  In general, researchers also work towards accelerated in vitro release assays which do tend to get a bit more complex since you should better understand the mechanism of release, and if it is degradation based, be able to mimic the profile observed in vivo (which is generally harder than you may think).  These accelerated in vitro release assays, if done correctly provide a tremendous amount of value but do take some investment to complete correctly.  The use of accelerated in vitro release should be included in this review.  Otherwise, very comprehensive review.

Author Response

Comment 1: In Section 4.10 Sterilization, the authors omitted the use of e-beam to sterilize the product.

Response: Included the use of e-beam to sterilize the product. Please refer to section 4.10, lines 1006-1008(as highlighted).

Comment 2: In section 5, on in vitro release, the authors describe the techniques as generally straightforward to implement, which may be true but the issue with in vitro release is that the assays take a long time to run,  since you generally want real-time release data (and these formulations are, after all, long-acting.  In general, researchers also work towards accelerated in vitro release assays which do tend to get a bit more complex since you should better understand the mechanism of release, and if it is degradation-based, be able to mimic the profile observed in vivo (which is generally harder than you may think).  These accelerated in vitro release assays, if done correctly provide a tremendous amount of value but do take some investment to complete correctly.  The use of accelerated in vitro release should be included in this review.  Otherwise, very comprehensive review.

Response: A sub-section on accelerated in vitro release studies for evaluating LAIs has been included in the manuscript. Please refer to section 5.5, Lines 1223 to 1292(as highlighted).

Reviewer 2 Report

Comments and Suggestions for Authors

The manuscript presents a comprehensive overview of long-acting injectable (LAI) formulations in veterinary medicine. It highlights various formulation strategies, emerging technologies, and the clinical significance of these formulations. While the topic is relevant and timely, the manuscript requires some revision before acceptance.

  1. The review primarily summarizes existing literature without providing novel insights. The discussion lacks a critical analysis of the recent advancements and potential breakthroughs in LAI technology.
  2. Consider adding a comparison with human LAI formulations to highlight unique challenges and opportunities in veterinary medicine.
  3. The review primarily summarizes existing literature without providing novel insights. The discussion lacks a critical analysis of the recent advancements and potential breakthroughs in LAI technology.
  4. Consider adding a comparison with human LAI formulations to highlight unique challenges and opportunities in veterinary medicine.
  5. The discussion on polymeric carriers (PLGA, PCL) lacks detailed mechanisms of drug release and degradation kinetics. Providing schematic illustrations or graphical representations would enhance clarity.
  6. More details on in vitro-in vivo correlation (IVIVC) are needed. Specific case studies or examples of successful IVIVC models would strengthen the discussion.
  7. While the manuscript briefly discusses biocompatibility, there is limited information on adverse reactions, immunogenicity, and regulatory challenges.
  8. Consider elaborating on safety profiles of various LAI excipients used in veterinary applications.
  9. Please ref this recent work, may be helpful , https://doi.org/10.1016/j.anifeedsci.2025.116259, https://doi.org/10.1016/j.aninu.2024.04.021

Reviewer 3 Report

Comments and Suggestions for Authors
  1. Line 14, please change Poly to poly.
  2. Paragraph 1 of 1.1 of Introduction please include pharmacokinetic and pharmacodynamic of medicine on animal of using pharmaceutical compounds with examples after review with supporting references.
  3.  Please add supporting refs in 2nd paragraph of Introduction 1.1.
  4. Before 1.1, Please start with how different between animals and human beings physiological body and acceptance to the medicine substances, since some drugs cannot use in the human but use in animal, please review and start at the first paragraph as Introduction. 
  5. For 1.2 please include route of administration of intra-periodontal pocket for long acting periodontitis treatment since current issue on this point is interesting and follow for literature review and include in the this review manuscript via different in situ forming drug delivery systems and you also mentioned in Line 189.
  6. For Atrigel® and other commercial products in 3.1, please review how to relate with veterinary application; please mention with examples, clinical evidence based and supporting refs. Please inform for in situ forming microparticle on periodontitis treatment and how any publication or clinical use with this type of dosage form. 
  7. Please address and review on previous publication related to in situ forming systems especially solvent exchange-induced in situ forming gel applied or used for periodontitis treatment and how their potential employing to veterinary care and use with schematic diagram of their phase transition.  Their benefit and limitation should be stated similar to Hydrogel system that you included.
  8. For 3.2, please review on application of active compounds from natural product in hydrogel application on veterinary.
  9. The in vivo, pharmacokinetic parameters or clinical evaluations of using hydrogel in veterinary should be included for hydrogel application with supporting refs.
  10. Line 338-339, Poly should be poly.
  11. ® please check for all content how to use superscript or not for consistency and how is this neccessary or nor to use capital letter for the name of them.
  12. Please check and review on ivermectin LAL of each dosage forms or topic in your review manuscript.
  13. Line 474, Procaine  should be procaine.
  14. Line 549, S. Enteritidis ??? Line 614 E. Gavini
  15. Please delete Fig. 2, since it is typical nanotechnology and you do not relate any thing to your animal application in this photo.
  16. Please include intra-pocket drug delivery into Table 3. 
  17. Please add rheological property into 4.3 with your review and supporting refs.
  18. For 4.3-4.9, please add the information directly review on these characteristics obtained on your dosage topic dosage form aimed to used in veterinary published or reported with supporting refs not just generalized writing.
  19. Before Conclusion, the physical and chemical stability should be another topic of review.
  20. The statement of using AI for writing should be claimed. 

Round 2

Reviewer 3 Report

Comments and Suggestions for Authors

Authors concern and revise this review manuscript properly and that improve progressively with high standard for publication.